# Quantum phases of hardcore bosons with repulsive dipolar density-density interactions on two-dimensional lattices

Jan Alexander Koziol[1*]⊙, Giovanna Morigi[2]⊙, and Kai Phillip Schmidt[1◇]⊙

**1** Department of Physics, Staudtstraße 7, Friedrich-Alexander-Universität
Erlangen-Nürnberg (FAU), Germany
**2** Theoretical Physics, Saarland University, Campus E2.6, D–66123 Saarbrücken,
Germany
⋆ jan.koziol@fau.de
◇ kai.phillip.schmidt@fau.de

September 12, 2024

## Abstract

We analyse the ground-state quantum phase diagram of hardcore Bosons interacting with repulsive dipolar potentials. The bosons dynamics is described by the extended-Bose-Hubbard Hamiltonian on a two-dimensional lattice. The ground state results from the interplay between the lattice geometry and the long-range interactions, which we account for by means of a classical spin mean-field approach limited by the size of the considered unit cells. This extended classical spin mean-field theory accounts for the long-range density-density interaction without truncation. We consider three different lattice geometries: square, honeycomb, and triangular. In the limit of zero hopping the ground state is always a devil's staircase of solid (gapped) phases. Such crystalline phases with broken translational symmetry are robust with respect to finite hopping amplitudes. At intermediate hopping amplitudes, these gapped phases melt, giving rise to various lattice supersolid phases, which can have exotic features with multiple sublattice densities. At sufficiently large hoppings the ground state is a superfluid. The stability of phases predicted by our approach is gauged by comparison to the known quantum phase diagrams of the Bose-Hubbard model with nearest-neighbour interactions as well as quantum Monte Carlo simulations for the dipolar case on the square and triangular lattice. Our results are of immediate relevance for experimental realisations of self-organised crystalline ordering patterns in analogue quantum simulators, e.g., with ultracold dipolar atoms in an optical lattice.

# 1  Introduction

Analogue quantum simulation poses a cornerstone of modern quantum technology [1–7]. One prominent quantum simulation platform are trapped ultracold atoms in optical lattice potentials [3, 8–12]. Thanks to the versatility of this experimental platform [13], these systems permit to shed light on quantum many-body phenomena [10, 13–15], as well as on the predictions of high-energy models [16, 17]. The experimental progress in cooling and trapping atoms and molecules with dipolar electric or magnetic moments [12, 18–20] enables experiments to realise extended Hubbard models with long-range interactions [21]. In the presence of a field, polarising the dipoles in the direction normal to the confinement plane, the effective potential is repulsive and scales with the distance $r$ between the particles as $1/r^3$. This opens novel perspectives for understanding the phase and dynamics of quantum systems with dipolar long-range interactions on two dimensional lattices.

The dynamics of the dipoles is usually modelled by the extended Hubbard model, where the effect of the long-range repulsion is described by a density-density interaction potential [22–30], truncated after the nearest-neighbour or the next-nearest-neighbour. For bosons, this term competes with tunnelling and contact interactions and is responsible for the onset of density modulations. The corresponding phases are denoted by charge density wave or solid phase when the phase is incompressible, and by lattice supersolid when the phase is superfluid [31, 32]. The most recent experiment by Su et al. [33] reported the onset

of solid phases in quantum magnetic gases of Erbium atoms for repulsive interactions in a two-dimensional square lattice.

In this work, we analyse theoretically how the interplay between lattice geometry and the full long-range interactions determines the quantum ground state of bosons. For this purpose we consider hard-core bosons with dipolar repulsive density-density interactions. We determine their phase diagram for three lattice geometries often discussed in the literature: (i) the square lattice, (ii) the honeycomb lattice, and (iii) the triangular lattice. With respect to the extensive literature on the subject, our approach is novel under several aspects. While often dipole-dipole interactions in lattice models are approximated by nearest-neighbor interactions [24, 28–30], here we do not perform any truncation but treat the full long-range density-density interactions by an appropriate resummation on finite unit-cells. This is implemented using the method we developed in Ref. [34], which extends the cluster-based classical spin mean-field calculation of Ref. [28, 35]. This allows us to perform calculations on all unit cells up to a feasible extent. We identify a number of features that are not captured by the nearest-neighbour truncation and unveil the effect of the interplay between the long-range dipolar interactions with the lattice geometry. In the limit of vanishing tunnelling, we show that the ground state is a devil's staircase of solid phases, which can be identified up to an arbitrary precision. This precision, in fact, is only limited by the size of the considered unit cells and the optimization scheme we employ. For finite tunnelling, we find solid and supersolid patterns, some of which have been reported by numerical studies based on advanced quantum Monte Carlo simulations [25, 36]. Differing from these works, we can avoid the limitation imposed by the constraints on the unit cell and thereby unveil a plethora of solid and supersolid structures which have not been reported before. Our results thus also provide an important benchmark and guidance for numerical programs, identifying the relevant unit cells, simulation geometries, and observables.

Finally, our predictions on hardcore bosons are directly relevant for XXZ quantum spin models in a longitudinal field with a long-range antiferromagnetic Ising interaction, as the Matsubara-Matsuda transformation [37] maps hardcore bosons to spin-1/2 degrees of freedom. Our work contributes to recent research work on how long-range interactions can give rise to new magnetic phases [34, 38–45] and alter the universality classes of quantum phase transitions [46–60]. To bridge the gap between the bosonic picture and the quantum spin model, we discuss the Hamiltonian and its ground states in the particle and the spin picture.

This paper is organised as follows. In Sec. 2 we introduce the model in the particle as well as the spin language. We emphasise the symmetries of the Hamiltonian and the nature of the occurring ground states in different parameter regimes. In Sec. 2.3 we define the three lattices geometries that are considered in this work (square, honeycomb and triangular lattice). In Sec. 3 we introduce the classical spin mean-field approach for long-range interacting systems. We discuss the classical spin approach in a general way to describe in detail how to treat long-range interactions in this framework. We then detail how we characterise ground states from the results of the mean-field calculation, and critically discuss the inherent limitations of the approach. In Sec. 4 we discuss the currently known quantum phase diagrams for the nearest-neighbour hardcore Bose-Hubbard model on the considered lattices. In Sec. 5 we present the mean-field quantum phase diagram for the hardcore Bose-Hubbard mode with repulsive dipolar density-density interactions on the bipartite square and honeycomb lattice. We focus on the devil's staircases of solids in Sec. 5.1. Further, we analyse the occurrence of supersolid phases and the experimental realisation of phases for the square lattice in Sec. 5.2.1 and for the honeycomb lattice in

Sec. 5.2.2. In Sec. 6 we present the results on the non-bipartite triangular lattice. We conclude our analysis in Sec. 7.

# 2 Hardcore bosons with repulsive dipolar density-density interactions

In this work, we consider hardcore bosons with repulsive dipolar density-density interactions confined on a two-dimensional lattice with a variable particle number. The lattice Hamiltonian for hardcore bosons with arbitrary algebraically decaying density-density interactions reads

$$H = -t \sum_{\langle i,j \rangle} \left( b_i^\dagger b_j + b_j^\dagger b_i \right) + \frac{V}{2} \sum_{i \neq j} \frac{1}{|\vec{r}_i - \vec{r}_j|^\alpha} n_i n_j - \mu \sum_i n_i \tag{1}$$

with hardcore bosonic creation (annihilation) operators $b_i^\dagger$ ($b_i$) and particle number operators $n_i$, where the index $i$ adresses operators at lattice site $\vec{r}_i = (r_{i,x}, r_{i,y})$. The parameters are the chemical potential $\mu$, the nearest-neighbour hopping amplitude $t$, the repulsion strength $V > 0$, and the algebraic decay exponent $\alpha$. In the rest of this work we consider the exponent $\alpha = 3$ of dipolar interactions.

The geometry of the lattice is taken to be (i) square, (ii) honeycomb, and (iii) triangular. Preceding studies of the model with nearest-neighbour interactions ($\alpha \to \infty$), showed that crystalline phases are present in all three lattice geometries [61–63], while lattice supersolid phases are stable only in the non-bipartite triangular lattice [61, 62]. A former study analysing the full long-range interations in the atomic limit $t = 0$ predicts a devil staircase of crystalline phases [34].

## 2.1 Description as a long-range XXZ quantum spin model

Hardcore bosonic operators can be bijectively mapped onto spin-1/2 operators using the Matsubara-Matsuda transformation [37]

$$n_i = \frac{1}{2} - S_i^z \qquad b_i^\dagger = S_i^- \qquad b_i = S_i^+ \tag{2}$$

converting the particle picture exactly into a spin model. By applying the Matsubara-Matsuda transformation to Eq. (1) we obtain an XXZ-model in a longitudinal field

$$\begin{aligned} H = &- 2t \sum_{\langle i,j \rangle} \left( S_i^x S_j^x + S_i^y S_j^y \right) + \frac{V}{2} \sum_{i \neq j} \frac{1}{|\vec{r}_i - \vec{r}_j|^\alpha} S_i^z S_j^z \\ &+ \left( \mu - \frac{V z_{\text{eff}}(\alpha)}{2} \right) \sum_i S_i^z + C(\mu, V, \alpha) \,. \end{aligned} \tag{3}$$

Here, $C(\mu, V, \alpha)$ is an irrelevant constant and $z_{\text{eff}}(\alpha)$ an effective coordination number,

$$C(\mu, V, \alpha) = \left( \frac{V z_{\text{eff}}(\alpha)}{8} - \frac{\mu}{2} \right) \sum_i \mathbb{1} \tag{4}$$

$$z_{\text{eff}}(\alpha) = \sum_{i=-\infty}^{\infty} \left( 1 - \delta_{0,i} \right) \frac{1}{|\vec{r}_i|^\alpha} \tag{5}$$

where for $\alpha = \infty$ the effective coordination number $z_{\text{eff}}(\infty)$ equals the coordination number of the nearest-neighbour model. In Eq. (3) the parameter $t$ now scales the ferromagnetic

nearest-neighbour XY-interaction while the parameters $\mu$, $V$ and $z_{\text{eff}}(\alpha)$ determine the longitudinal field. Further, we obtain an antiferromagnetic long-range Ising interaction with amplitude $V$ and decay exponent $\alpha$.

## 2.2 Symmetries

The Hamiltonian (1) is particle-hole symmetric for any value of the exponent $\alpha$. As a consequence, the resulting quantum phase diagrams are also particle-hole symmetric around the line where $\mu/V = z_{\text{eff}}(\alpha)/2$. This can be shown by exchanging particle $b_i, b_i^\dagger, n_i$ and hole $\tilde{b}_i, \tilde{b}_i^\dagger, \tilde{n}_i$ operators in Eq. (1) as $b_i \leftrightarrow \tilde{b}_i^\dagger$ (with the hardcore bosonic commutation relation $[b_i, b_j^\dagger] = \delta_{i,j}(1 - 2n_i)$ follows $n_i = 1 - \tilde{n}_i$) and realising that the configuration of holes at $V z_{\text{eff}}(\alpha) - \mu$ corresponds to a configuration of particles at $\mu$. This implies that for $t = 0$ the filled phase is reached at $\mu/V = z_{\text{eff}}(\alpha)$. In the magnetic picture this particle-hole symmetry corresponds to a spin inversion $\vec{S}_i \leftrightarrow -\vec{S}_i$ about the symmetry line where the longitudinal field vanishes, $\mu/V = z_{\text{eff}}(\alpha)/2$ [34].

The model of interest is invariant under $U(1)$ transformations of the form $b_i \to b_i e^{i\psi}$ with $\psi \in \mathbb{R}$. This implies that the Hamiltonian (1) is particle conserving and equivalently Hamiltonian (3) conserves the $z$-magnetisation $M^z = \sum_i S_i^z$.

## 2.3 Definition of the considered lattices and unit cells

We now turn to the lattice geometries considered in this work. Figure 1 illustrates the square, the honeycomb, and the triangular lattice. All considered unit cells with the respective resummed couplings can be found in Ref. [64]. The sets of considered translational vectors Eqs. (6), (7) and (8) are chosen to balance the trade-off between the number of unit cells, the computational effort to determine the resummed couplings, and the time for the numerical optimization on each unit cell.

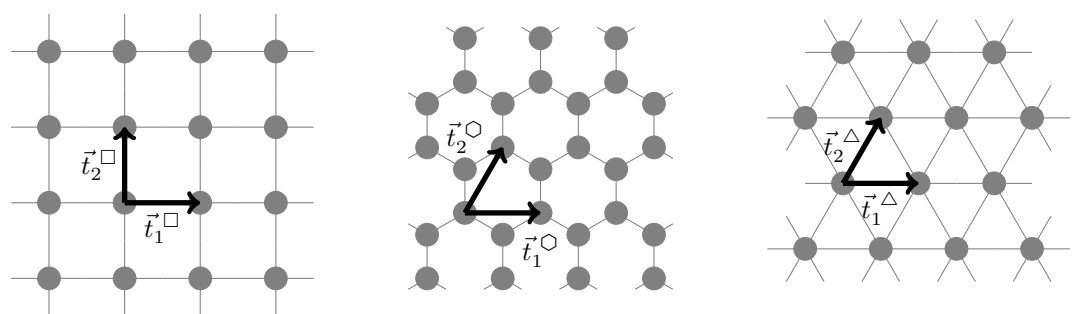

Figure 1: Illustration of the the square, honeycomb and triangular lattice (from left to right). The corresponding primitive lattice vectors $\vec{t}_i^\Lambda$ (with $\Lambda \in \{\square, \hexagon, \triangle\}$) are shown.

### 2.3.1 Square lattice

The square lattice is a two-dimensional bipartite [1] Bravais lattice with primitive translation vectors $\vec{t}_1^\square = (1, 0)^{\text{T}}$ and $\vec{t}_2^\square = (0, 1)^{\text{T}}$ (see. Fig 1). For the procedure described in Sec. 3

---

[1]A lattice is called bipartite if there exists a decomposition of the lattice sites into two disjoint sets $A$ and $B$ such that every lattice site in set $A$ is only neighbouring sites from set $B$ and vice versa.

we use all distinct unit cells with translation vectors $\vec{T}_1, \vec{T}_2 \in \mathcal{A}_6^\square$ out of the set [34]

$$\mathcal{A}_6^\square = \{i\vec{t}_1^\square + j\vec{t}_2^\square | i \in \{-6, ..., 6\}, j \in \{-6, ..., 6\}\} . \tag{6}$$

The particle-hole symmetry line for the nearest-neighbour model on the square lattice is at $\mu/V = z^\square/2 = 2$ and for the model with dipolar interactions at $\mu/V = z_{\text{eff}}^\square(3)/2 = 4.51681084$.

### 2.3.2 Honeycomb lattice

The honeycomb lattice is a two-dimensional bipartite lattice. It can be understood as a triangular lattice with two sites per unit cell $\vec{\delta}_1^\bigcirc = (0,0)^\text{T}$ and $\vec{\delta}_2^\bigcirc = (0,1)^\text{T}$ with primitive translation vectors $\vec{t}_1^\bigcirc = (\sqrt{3}, 0)^\text{T}$ and $t_2^\bigcirc = (\sqrt{3}/2, 3/2)^\text{T}$ (see. Fig 1). We use all distinct unit cells with translation vectors $\vec{T}_1$ and $\vec{T}_2$ out of the set [34]

$$\mathcal{B}_6^\bigcirc = \{i\vec{t}_1^\bigcirc + j\vec{t}_2^\bigcirc | i \in \{-6, ..., 6\}, j \in \{\min(-6 - i, -6), ..., \min(6 - i, 6)\}\} . \tag{7}$$

The particle-hole symmetry line for the nearest-neighbour model on the honeycomb lattice is at $\mu/V = z^\bigcirc/2 = 1.5$ and for the model with dipolar interactions at $\mu/V = z_{\text{eff}}^\bigcirc(3)/2 = 3.28942596$.

### 2.3.3 Triangular lattice

The triangular lattice is a two-dimensional non-bipartite Bravais lattice with primitive translation vectors $\vec{t}_1^\triangle = (1,0)^\text{T}$ and $\vec{t}_2^\triangle = (1/2, \sqrt{3}/2)^\text{T}$ (see. Fig 1). We use all distinct unit cells with translation vectors $\vec{T}_1$ and $\vec{T}_2$ out of the set [34]

$$\mathcal{B}_6^\triangle = \{i\vec{t}_1^\triangle + j\vec{t}_2^\triangle | i \in \{-6, ..., 6\}, j \in \{\min(-6 - i, -6), ..., \min(6 - i, 6)\}\} . \tag{8}$$

The particle-hole symmetry line for the nearest-neighbour model on the triangular lattice is at $\mu/V = z^\triangle/2 = 3$ and for the model with dipolar interactions at $\mu/V = z_{\text{eff}}^\triangle(3)/2 = 5.517087866$.

## 2.4 Quantum phases

In this section we discuss all relevant phases of the ground state of Hamiltonian (1) and their salient properties, see also Table 1 for a summary.

In the limit of large chemical potentials $\mu \gg t, V$ the ground state of the system is given by a phase with a filling of $n_i = 1$ at every site [65]. The corresponding magnetic picture is a perfect magnetic alignment in $S_i^z = -1/2$ direction (see Eq. (2)). This phase is symmetric with respect to all symmetries of the Hamiltonian. It has a finite elementary excitation gap, it is incompressible (the density is constant in $\mu$), and it is stable against finite quantum fluctuations induced by the hopping $t$ [65]. The fully filled or fully polarised state is an exact eigenstate of the Hamiltonian and realises the ground state anywhere within this phase.

Similarly, at negative chemical potentials $\mu$, the ground state of the system is given by a phase with a filling of $n_i = 0$ at every site corresponding to a unit filling of holes [65]. The corresponding magnetic picture is a perfect alignment in $S_i^z = 1/2$ direction (see Eq. (2)). Due to the particle-hole symmetry, the properties of the empty phase are the same as of the completely filled phase [65].

In the limit of large $t \gg \mu, V$, the kinetic energy of the particles is dominant and the ground state of the system is a superfluid phase [65]. In the superfluid phase the expectation value of the annihilation operator is non-zero $\langle b_i \rangle \neq 0$ [65], signalling the spontaneous breaking of the $U(1)$ symmetry of the Hamiltonian. We denote the corresponding order by off-diagonal long-range order [66]. In the magnetic language, the superfluid behaviour translates to a finite transverse XY-magnetisation breaking the $U(1)$ symmetry of the

Hamiltonian (3) with respect to rotations around the $z$-axis. Note, this phase does not break the discrete translational symmetry of the Hamiltonian, therefore there are no density modulations in the system.

Ground states with periodic density modulation break the discrete translational symmetry of the Hamiltonian and possess diagonal long-range order [66]. Diagonal long-range order is characterised by a periodicity in the local particle densities which is quantified using the static structure factor at the momentum $\vec{k} = \vec{k}_{\text{order}}$ associated with the periodicity of the pattern

$$S(\vec{k}) = \frac{1}{N^2} \sum_{l,m} \exp(\mathrm{i}\vec{k} \cdot (\vec{r}_l - \vec{r}_m)) \langle n_l n_m \rangle. \tag{9}$$

Phases with diagonal long-range order but no superfluidity are here dubbed solid phases. These phases can occur with different fractions $f = n/m$ of occupied sites, with $m, n$ natural numbers prime to each other. In order to characterise them, we name them $f = n/m$ solids. These solid phases are commensurate. We will show that, in the limit of vanishing hopping $t = 0$, they form a devil staircase as a function of the chemical potential. In the magnetic picture these phases correspond to a magnetic order in $z$-direction with unit cells composed by multiple sites and no XY-magnetisation. These phases are gapped and therefore stable against quantum fluctuations at finite $t$ [34].

In addition to the phases discussed so far, we will also report lattice supersolid phases, namely, phases displaying both diagonal and off-diagonal long-range order. These phases break the discrete translational symmetry of the Hamiltonian and the $U(1)$ symmetry. In the magnetic picture, this translates to a magnetic order in $z$-direction and a finite XY-magnetisation. We characterise a simple supersolid phase with two sublattices at a different mean occupation in the same spirit as the solids. For instance, an $n/m$ supersolid is characterised by a fraction of $n/m$ sites having a certain mean occupation and $(m - n)/m$ with a different one. We will demonstrate that in the presence of long-range interactions also supersolid phases with complex sublattice structures occur. We name them according to the fractions of lattice sites at each mean occupation. An example would be the 1/2-1/12-5/12 supersolid discussed in Fig. 6. In order to identify supersolids with a complex sublattice structure, a careful choice of an order parameter is required, see Sec. 5.2.1.

In the following we will drop the specification „lattice" when referring to the supersolid phases.

| Quantum phase | DLRO | OLRO | Gap $\Delta$ | $S(\vec{k}_{\text{order}})$ | $\langle b_i \rangle$ |
|---|---|---|---|---|---|
| empty solid | no | no | $> 0$ | $= 0$ | $= 0$ |
| fully filled solid | no | no | $> 0$ | $= 0$ | $= 0$ |
| $f = n/m$ solid | yes | no | $> 0$ | $\neq 0$ | $= 0$ |
| $n/m$ supersolid | yes | yes | $= 0$ | $\neq 0$ | $\neq 0$ |
| complex supersolids | yes | yes | $= 0$ | $\neq 0$ | $\neq 0$ |
| superfluid | no | yes | $= 0$ | $= 0$ | $\neq 0$ |

Table 1: Summary of the occurring quantum phases in the considered hard-core Bose-Hubbard model (1) with repulsive dipolar density-density interactions. DLRO (OLRO) stands for (off)diagonal long-range order. The gap $\Delta$ refers to the elementary excitation gap. The ordering wave-vector $\vec{k}_{\text{order}}$ is determined by the periodicity of the density pattern. Complex supersolids refer to supersolids with a sublattice structure with more than two sublattices

.

# 3 Classical spin mean-field calculations with long-range interactions

In this work, we extend the procedure developed in Ref. [34] to perform classical spin mean-field calculations [35] for hardcore bosonic particles with algebraically decaying density-density interactions by including the effect of finite hoppings. The first step of the classical spin mean-field approach is performed by mapping the hardcore bosonic operators onto spin-1/2 operators using the Matsubara-Matsuda transformation Eq. (2) [35,37]. The spins are treated as classical vectors $\vec{S}_i$ of length $|S_i| = 1/2$ [35,67]. For a spin $\vec{S}_i$ the vector can be parametrised on a sphere as

$$\vec{S}_i = \begin{pmatrix} \bar{S}_i^x \\ \bar{S}_i^y \\ \bar{S}_i^z \end{pmatrix} = \frac{1}{2} \begin{pmatrix} \sin(\theta_i)\cos(\phi_i) \\ \sin(\theta_i)\sin(\phi_i) \\ \cos(\theta_i) \end{pmatrix} \tag{10}$$

with $\phi_i \in [0, 2\pi)$ and $\theta_i \in [0, \pi]$. Within this approximation the ground state is given by a classical arrangement of spins that minimises the classical spin Hamiltonian [35]

$$\bar{H} = -2t \sum_{\langle i,j \rangle} \left( \bar{S}_i^x \bar{S}_j^x + \bar{S}_i^y \bar{S}_j^y \right) + \frac{V}{2} \sum_{i \neq j} \frac{1}{|\vec{r}_i - \vec{r}_j|^\alpha} \bar{S}_i^z \bar{S}_j^z$$
$$+ \left( \mu - \frac{V z_{\text{eff}}(\alpha)}{2} \right) \sum_i \bar{S}_i^z + C(\mu, V, \alpha) . \tag{11}$$

The minimization is then performed numerically on unit cells following the ideas described in Ref. [34]. The underlying spirit of the calculations on finite unit cells performed in this work can be summarised as follows:

1. Consider systematically all possible unit cells up to a certain extent.

2. Treat the long-range interaction on each unit cell using appropriately resummed interactions.

3. Determine the optimal pattern on each unit cell for a given set of parameters.

4. Compare the energies per site between each unit cell to determine the overall optimal pattern minimizing the energy.

## 3.1 Resummed couplings for periodic ordering patterns

The core observation of our method is that one can rewrite a long-range density-density interaction for a periodic pattern with an $N$-site unit cell and translational vectors $\vec{T}_1$ and $\vec{T}_2$

$$\frac{V}{2} \sum_{i \neq j} \frac{1}{|\vec{r}_i - \vec{r}_j|^\alpha} n_i n_j = \frac{1}{2} \sum_{i=1}^N \tilde{V}_{i,i}^{\infty,\alpha} n_i n_i + \frac{1}{2} \sum_{\substack{i,j=1 \\ i \neq j}}^N \tilde{V}_{i,j}^{\infty,\alpha} n_i n_j \tag{12}$$

into sums over the unit cell of the pattern using appropriately resummed couplings

$$\tilde{V}_{i,j}^{\infty,\alpha} = V \sum_{l=-\infty}^{\infty} \sum_{k=-\infty}^{\infty} \frac{1}{|\vec{r}_i - \vec{r}_j + l\vec{T}_1 - k\vec{T}_2|^\alpha} \tag{13}$$

$$\tilde{V}_{i,i}^{\infty,\alpha} = V \sum_{l=-\infty}^{\infty} \sum_{k=-\infty}^{\infty} \frac{(1-\delta_{l,k})}{|l\vec{T}_1 - k\vec{T}_2|^{\alpha}} \tag{14}$$

with $\delta_{k,l}$ being the Kronecker delta. A detailed description on how to systematically determine all unit cells up to a given extent and how to determine the resummed couplings by brute force real-space summation can be found in Ref. [34].

During the refereeing process of the manuscript, we became aware of the fact that the resummed couplings can be represented using the Epstein $\zeta$-function [68, 69]. Recently an efficient numerical implementation of the Epstein $\zeta$-function was introduced [70–73]. This approach outperforms the brute force summation in speed and accuracy of the results and enables the evaluation of resummed couplings up to machine precision without any effort. We have checked that our previously (by brute force summation) obtained resummed couplings used in this publication and in [34] (see [64] and [74]) are within a tolerance of $10^{-10}$ to these "perfect" resummed couplings. For further projects we recommend using an efficient implementation of the Epstein $\zeta$-function e. g. [75].

## 3.2 Mean-field calculations with long-range interactions

In the following, we present how one can use the resummed couplings in order to make long-range interactions accessible to mean-field calculations requiring an underlying cluster. In the spirit of our approach, we are assuming that the optimal arrangement of angles $\theta_i$ and $\phi_i$ is periodic with some unit cell.

Using resummed long-range interactions, the Hamiltonian in Eq. (11) on an $N$-site unit cell with translational vectors $\vec{T}_1$ and $\vec{T}_2$ reads in the classical spin approximation

$$\bar{H} = -2t \sum_{i,j=1}^{N} \chi_{\langle i,j \rangle} \left( \bar{S}_i^x \bar{S}_j^x + \bar{S}_i^y \bar{S}_j^y \right) + \frac{1}{2} \sum_{\substack{i,j=1 \\ i \neq j}}^{N} \tilde{V}_{i,j}^{\infty,\alpha} \bar{S}_i^z \bar{S}_j^z$$
$$+ \frac{1}{2} \sum_{i}^{N} \tilde{V}_{i,i}^{\infty,\alpha} \bar{S}_i^z \bar{S}_i^z + \left( \mu - \frac{V z_{\text{eff}(\alpha)}}{2} \right) \sum_{i}^{N} \bar{S}_i^z + c(\mu, V, \alpha) \tag{15}$$

with $\chi_{\langle i,j \rangle}$ being an indicator function which is one if $i$ and $j$ are nearest neighbours including periodic boundary conditions and is zero in all other cases. The value of the irrelevant constant is $c(\mu, V, \alpha)$ is equal to $C(\mu, V, \alpha)$ divided by the number of unit cells. Inserting the parametrisation of the classical vector on the sphere, we derive the following energy expression

$$E(t, V, \mu; \phi_1, ..., \theta_N)/N = -\frac{2t}{4N} \sum_{i,j=1}^{N} \sin(\theta_i) \sin(\theta_j) + \frac{1}{8N} \sum_{i=1}^{N} \tilde{V}_{i,i}^{\infty,\alpha} \cos^2(\theta_i) \tag{16}$$
$$+ \frac{1}{8N} \sum_{\substack{i,j=1 \\ i \neq j}}^{N} \tilde{V}_{i,j}^{\infty,\alpha} \cos(\theta_i) \cos(\theta_j) + \frac{(\mu - \frac{1}{2} V z_{\text{eff}}(\alpha))}{2N} \sum_{i} \cos(\theta_i)$$

that shall be minimised in order to find the optimal arrangement of angles $\{\phi_1, \theta_1, ..., \phi_N, \theta_N\}$. Note, that this expression is independent of the $\phi_i$ angles. In fact, one can show from Eq. (15) that the energy of a state is solely dependent on angle differences $\phi_i - \phi_j$. One can then further see that in order to minimise the energy the angle differences need to be zero. Therefore, we can w. l. o. g. set $\phi_j = 0$ with $j \in \{1, ..., N\}$ to search for the optimal $\theta_i$ values.

Following the strategy identified in Ref. [34], we search for the optimal pattern on each unit cell for a given set of parameters and compare the energies per site between each unit cell to determine the overall optimal pattern. For the global optimizations, we use the locally biased variant [76] of the dividing rectangles global optimization algorithm [77]. In addition, we apply the local low-storage Broyden-Fletcher-Goldfarb-Shanno algorithm [78–83] for numerous starting configurations at uniform densities as well as for solid and two-sublattice supersolid states.

## 3.3   Characterisation of phases

Having determined the energetically optimal arrangement of angles, the next step is to characterise the type of phase in order to determine a ground-state phase diagram. The result of the optimization is an estimate for the ground-state energy per site in the mean-field framework and a corresponding configuration of angles $\{\theta_1, ..., \theta_N\}$ with the number of sites $N$ in the unit cell of the ground state. Making use of the correspondence between spins and occupations, we can convert the $\theta$-angles to local densities $n_i = (1 - \cos(\theta))/2$. The goal is to translate the general characterisation of the different quantum phases in Tab. 1 to the classical spin approach framework. Using solely the $\theta$-angles we classify the phases as follows:

- If $\theta_j = 0$ with $j = \{1, ..., N\}$ the ground state realises the empty state. Analogously, if $\theta_j = \pi$ with $j = \{1, ..., N\}$ the ground state realises unit filling.

- If $\theta_j = \theta_{\mathrm{SF}}$ with $\theta_{\mathrm{SF}} \notin \{0, \pi\}$ and $j = \{1, ..., N\}$ the ground state is a superfluid state with a mean local density of $\rho = (1 - \cos(\theta_{\mathrm{SF}}))/2$.

- If a fraction $f = n/m$ of angles is $\pi$ and the other angles are 0, the system realises a $f = n/m$ solid according to the number of occupied sites.

- The simplest form of a supersolid has a fraction $f = n/m$ of sites having an angle $\theta_{\mathrm{SS},A}$ and the rest having an angle $\theta_{\mathrm{SS},B}$. We require w. l. o. g. that $\pi > \theta_{\mathrm{SS},A} > \theta_{\mathrm{SS},B} > 0$ and call the sites with an angle $\theta_{\mathrm{SS},A}$ ($\theta_{\mathrm{SS},B}$) majority (minority) sites as they have a major (minor) onsite density. We denote these two-flavour supersolids by „$n/m$ supersolids" reflecting the number of majority sites.

- The last type of phases relevant for the considerations in this work are supersolid phases with three and four different densities $0 < \theta_{\mathrm{SS},A} < \theta_{\mathrm{SS},B} < \theta_{\mathrm{SS},C} (< \theta_{\mathrm{SS},D}) < \pi$. We name these supersolids according to the fractions of sites with a certain angle from the smallest to the largest angle. In principle there can be supersolid phases with more then four sublattices. However, it is not feasible to auto-characterise them due to numerical noise on the optimised angles.

Finally, in view of the results of studies of the nearest-neighbour models and quantum Monte Carlo simulations of the dipolar model on the square lattice, we see no reason that pair condensate phases [84,85] occur in the long-range phase diagram. Therefore, we expect the ansatz here described to capture all relevant phases.

With this ansatz, the ground state is now completely characterised by the set of the angles $\{\theta_1, ..., \theta_N\}$. This allows us to calculate other observables. One simple example is the mean density, which takes the form

$$\rho = \frac{1}{N} \sum_{i=1}^{N} (1 - \cos(\theta_i))/2 \ . \tag{17}$$

The framework of our method shall allow us to detect all relevant phases characterizing the ground state of the mean-field model. Note, however, it is in some cases difficult to perform the identification according to the scheme presented above, as the precise systematic comparison of floating point numbers determined from the numerical optimization procedures requires the introduction of tolerances and leads in some cases to inconclusive results.

## 3.4 Limitations of the classical mean-field approach

From the application of the classical spin approximation to two-dimensional systems with finite-range interactions and comparison with numerical methods treating the quantum fluctuations in their full extent, it has been established, that the method captures occurring phases in a good qualitative manner [35]. The classical approximation underestimates the effect of quantum fluctuations, therefore it predicts that solid and supersolid phases are stable in a larger region of the phase diagram than predicted by numerical calculations, which accurately take quantum fluctuations into account [62]. Nevertheless, some features of phase diagrams such as the boundary between the empty and the superfluid phase are captured exactly by the classical spin approach [61,62]. The modification due to the long-range density-density repulsion affects only the diagonal part of the Hamiltonian in the chosen basis. In the atomic limit, where the Hamiltonian is diagonal, it is treated exactly by our approach [34]. Therefore, we expect that our predictions will be limited by the underestimation of quantum fluctuations in a similar fashion it occurs in finite-range interacting systems. In this work, we additionally compare our results to quantum Monte Carlo studies on finite systems, which systematically take quantum fluctuations into account. In general, the strength of the quantum fluctuations will decrease with increasing coordination number, system dimension, or spin quantum number. Despite its limitations, the classical spin model predicted the magnetisation plateaux of the Shastry-Sutherland spin model which is realised in the frustrated quantum magnet $SrCu_2(BO_3)_2$ compound in former studies [35, 86–88].

## 4 Quantum phase diagram for nearest-neighbour interactions

In order to benchmark the effect of the full dipolar density-density interaction on the phase diagram of the Bose-Hubbard model in Eq. (1) and to demonstrate the capabilities of the classical spin approach, we first discuss the phase diagrams when the interaction in Eq. (1) is truncated to the nearest-neighbours. Figure 2 summarises results of preceding stochastic series expansion quantum Monte Carlo studies on the square [26, 27, 89], honeycomb [63], and triangular [62] lattice. These Monte Carlo results fully capture the effects of quantum fluctuations. In the same figure, we also present the results from the classical spin approach for the corresponding model, thereby illustrating its capabilities and drawbacks in a pictorial way.

We first observe that all diagrams are symmetric about the line $\mu/V = z/2$ (with $z$ being the coordination number of the lattice). This is a consequence of the particle-hole symmetry discussed in Sec. 2.2, and becomes particularly evident in the atomic limit, for $t = 0$. The empty solid is present at $\mu < 0$, where there is no energetic benefit of having particles in the system. Correspondingly, the fully filled phase is found for $\mu/V > z$.

The difference between the geometries becomes most prominent for values of $\mu$ between these two values. The two bipartite lattices have a rather similar phase diagram: Here, we find a solid phase with a fraction of $f = 1/2$ sites occupied. This ordering pattern is a direct consequence of the bipartition as the density-density repulsion always connects

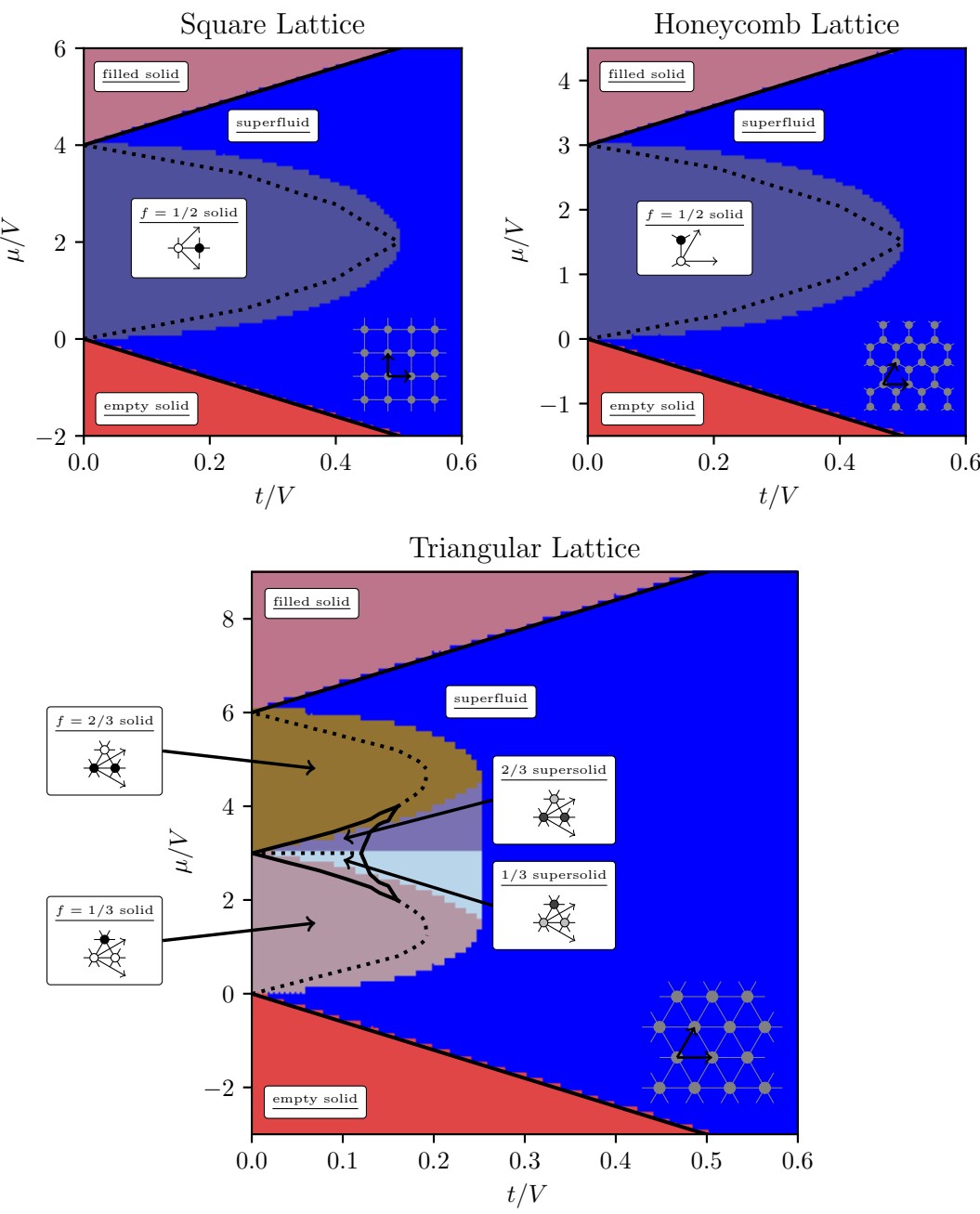

Figure 2: Quantum phase diagrams of the hardcore Bose-Hubbard model with repulsive nearest-neighbour density-density interactions (Eq. (1) with $\alpha = \infty$) for the square lattice (upper left panel), honeycomb lattice (upper right panel) and triangular lattice (lower panel). The dashed (solid) lines indicate first (second) order phase transition lines between phases calculated using numerical quantum Monte Carlo simulations. The data of these lines is extracted for the square lattice from Ref. [27], for the honeycomb lattice from Ref. [63], and for the triangular lattice from Ref. [62]. The underlying colour map represents the corresponding results from the classical spin approach described in Sec. 3. The insets display the structure of the solid and supersolid phases: the colors of the sites represent the mean occupation according to a grey scale where the filled sites are black and the empty white. The arrows illustrate the translational vectors of the depicted unit cells.

sites between the two lattice partitions. Instead, for the triangular lattice we find two solid phases separated by the particle-hole symmetry line: the $f = 1/3$, where no neighbouring sites are simultaneously occupied, and the $f = 2/3$, which is the particle-hole symmetric counterpart. Moreover, supersolid phases are found only in the non-bipartite triangular lattice: These are the 1/3 and 2/3 supersolid phases and are localised on the inner sides of the lobes where the corresponding solid phase is stable.

All the solid phases are gapped. When increasing the hopping $t/V$, there occurs at some point a transition to a phase with off-diagonal long-range order. First-order phase transitions separate two phases with different diagonal long-range order. This is the case for the phase transition on the square and honeycomb lattice separating the $f = 1/2$ solid and the superfluid phase. Other examples are found for the triangular lattice, such as the transition between the $f = 1/3$ solid and the superfluid (and correspondingly the one separating the $f = 2/3$ solid and the superfluid), as well as the supersolid-supersolid transition at the particle-hole symmetry line. Instead, the phase transitions are of second order if diagonal long-range order is preserved. This is demonstrated in the phase transition separating the empty/filled phase and the superfluid phase. It is also verified for the phase transition between the $f = 1/3$ solid and 1/3 supersolid (and correspondingly for $f = 2/3$).

The comparison between the predictions of the quantum Monte Carlo and the classical spin mean-field theory shows that the two methods are in qualitative agreement. The agreement is quantitative for the prediction of the size of the empty and of the filled solid. The transition line can be determined in first-order perturbation theory [62]. The vacuum and the one-particle states of the empty/filled phase are not dressed by quantum fluctuations. This lack of quantum fluctuations in the state makes the classical spin approach a viable method to exactly determine the position of the transition line. As far as it concerns the size of the other solid phases, the classical approximation captures correctly the tip of the $f = 1/2$ solid lobes of the square and honeycomb lattice at $\mu/V = z/2$ and $t/V = 1/2$. This point corresponds to the so-called Heisenberg point, where the Hamiltonian in spin picture Eq. (3) becomes the $SU(2)$-symmetric Heisenberg Hamiltonian. For $t/V = 1/2$, at $\mu/V = z/2$, the Heisenberg Hamiltonian with a gapless ground state is recovered [90–93] while for $t/V < 1/2$ the phase is the ground state of a gapped easy-axis magnet in the absence of a field [90–93]. The transition at the Heisenberg point is captured correctly by the classical spin approach since it originates from the symmetry change from an easy-axis to a full rotationally symmetric magnetic interaction. This change is occurring for the same parameter value in the classical spin approach, as well as, in the full quantum mechanical problem. Interestingly, when regarding the full quantum mechanical problem the gap $\Delta(t/V \to 1/2)$ closes with a mean-field gap exponent [93, 94].

Apart for the Heisenberg point, the classical spin approach visibly underestimates the effect of quantum fluctuations and therefore overestimates the size of the phases with diagonal long-range order. The largest deviation is found in the prediction of the size of the supersolid phases in the triangular lattice. It is believed that the phase diagram on the triangular lattice with the frustrated nearest-neighbour interaction is changed the most since "frustration enhances the effects of quantum fluctuations [61]". Long-range interactions can be understood as a hierarchy of competing constraints. Therefore, it seems reasonable that the phases we observe in the classical spin approach for long-range interacting systems will be strongly impacted by quantum fluctuations.

Despite these discrepancies, the existence of phases predicted by the classical spin approach is also confirmed by large-scale numerical calculations [26, 27, 62, 63, 89]. The phase diagrams presented in Fig. 2 agree qualitatively.

Next, we analyse the effect of power-law interactions on the resulting phases. Following the qualitatively different shape of the nearest-neighbour phase diagrams depending on

whether the lattice is bipartite or not, we structure our discussion first analysing bipartite lattices and then separately the triangular lattice.

# 5 Quantum phase diagrams of long-range interacting bosons: bipartite lattices

In this section, we analyse quantum phase diagrams of the hardcore Bose-Hubbard model with repulsive dipolar density-density interactions on the bipartite square and honeycomb lattice. We refer to Ref. [64] for the data on the classical spin approach ground states $\{\theta_1, ..., \theta_N\}$ resulting from the numerical optimization procedure. This data was used to create Figs. 5-8.

## 5.1 Atomic limit: The devil's staircase

We first discuss the atomic limit $t/V = 0$. In this limit the employed procedure is equivalent to the approach described in Ref. [34] and the classical spin approach becomes exact as long as the ground states fit onto the considered unit cells.

There are two features that are common to the nearest-neighbour case and to the dipolar interactions: the empty solid for $\mu < 0$, the particle-hole symmetric filled solid, and the $f = 1/2$ chequerboard solid phase around the particle-hole symmetry line. In the nearest-neighbour case the $f = 1/2$ solid phase is the only one besides the two trivial solids and it extends all the way down to $\mu/V = 0$ at the $t/V = 0$ line. For the dipolar interactions the lobe of the $f = 1/2$ solid terminates at around $\mu/V \cong 3.197$ for the square lattice and around $\mu/V \cong 2.18388$ for the honeycomb lattice.

Figure 3 displays the fractional solid phases as a function of $\mu$. These are reported in the interval between $\mu = 0$ and the value at the symmetry line, which corresponds to $\mu/V \approx 4.51681084$ for the square lattice and to $\mu/V \approx 3.28942596$ for the honeycomb lattice.

The ground state appears to possess fractal properties with the characteristic of a devil's staircase: as a function of $\mu$ we observe plateaux where a certain fractional solid $f$ is stable. However, the number of fractional states increases by increasing the size of the unit cell while taking a finer grid in $\mu/V$ (see Ref. [34]). Some of the configurations are illustrated in the lower part of Fig. 3.

In Fig. 4 we compare the widths of the plateaux in $\mu/V$ for certain fractional solids $f$. In general, the plateau width of fractional solids of the honeycomb lattice is substantially narrower than the corresponding ones in the square lattice, even though in some special cases the opposite occurs. This seems to be related to the size and symmetry of the unit cell that characterises the corresponding pattern. In particular, if the number of sites is the same, then the plateaus on the honeycomb lattice have a larger extent than the ones on the square lattice. Note, the translational vectors of these patterns have an hexagonal symmetry, see patterns in Fig. 3. On the other hand, the width of the plateaux at $f = 1/3$, $f = 1/4$, and $f = 1/5$ are narrower in the honeycomb lattice, and correspond to patterns that have substantially larger unit cells.

## 5.2 Quantum phase diagrams

Here, we analyse the quantum phase diagram of the square and honeycomb lattice as a function of the tunnelling amplitude $t$ and of the chemical potential $\mu$ in units of the interactions strength $V$. Figure 5 shows the phase diagram below the particle-hole symmetry line at $\mu/V = z_{\text{eff}}(3)/2 \cong 4.51681084$. The upper half is directly related using the particle-hole

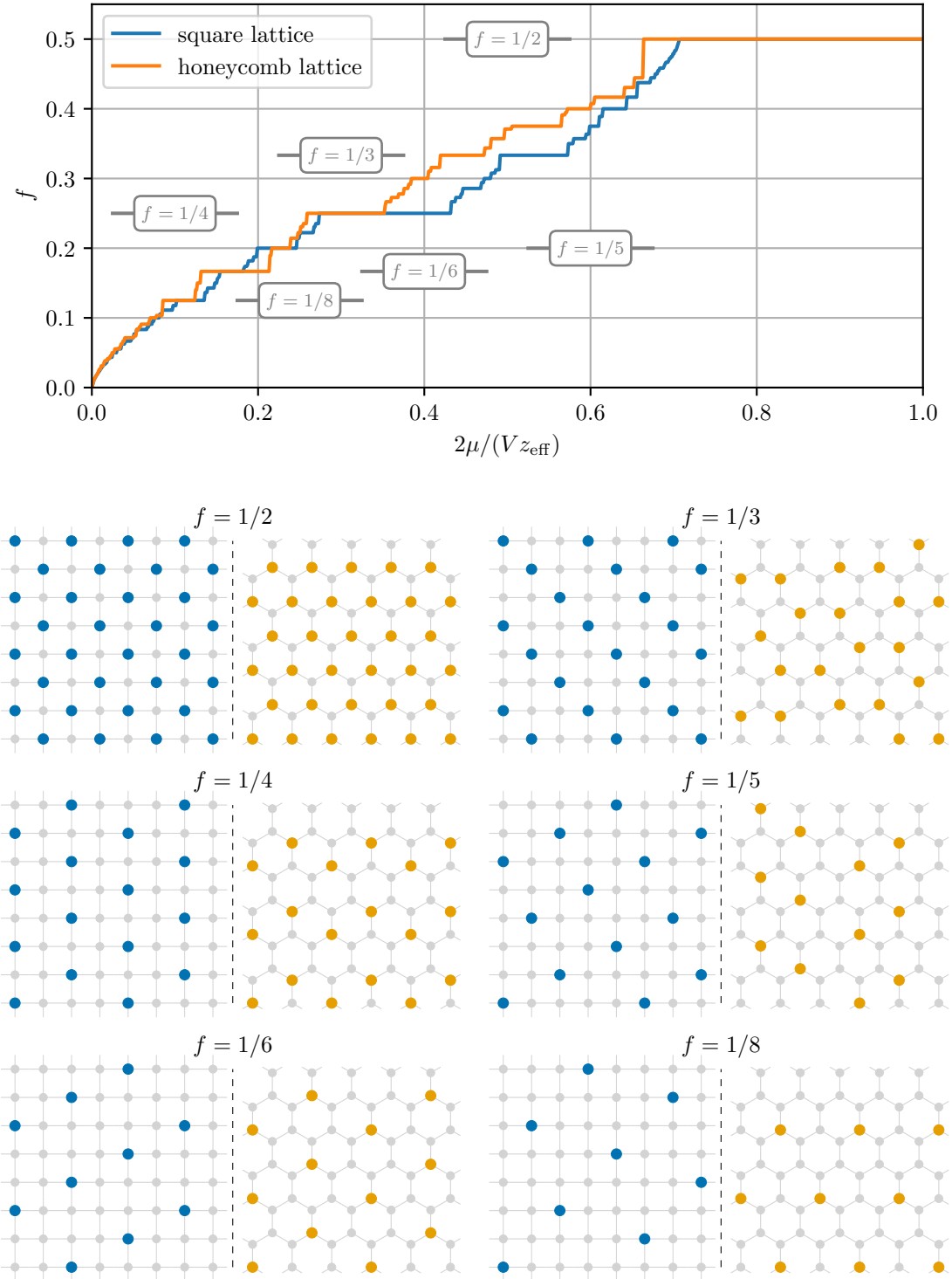

Figure 3: Devil's staircase of solid phases with filling $f$ of the hardcore Bose-Hubbard model with repulsive dipolar long-range density-density interactions at $t/V = 0$. The staircases in $\mu/V$ of the square and honeycomb lattice are normalised to the particle-hole symmetry line in order to compare both lattices on an equal footing. In the lower panels, we depict some solid phases of the staircase. Blue (orange) circles indicate occupied sites on the square (honeycomb) lattice and light grey circles indicate empty sites.

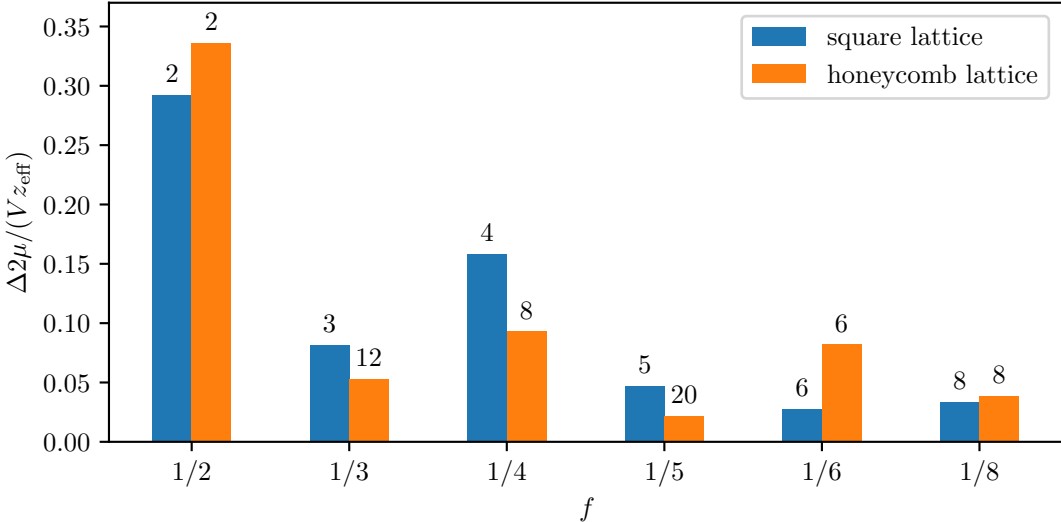

Figure 4: Width of the plateaux of Fig. 3 as a function of some selected fractional solids $f$. Blue (orange) labels the square (honeycomb) lattice. The numbers above the bars denote the number of sites characterising the elementary unit cell of the corresponding pattern.

symmetry of the Hamiltonian. The phase diagram of the honeycomb lattice is presented in Fig. 8, where the particle-hole symmetry line is at $\mu/V = z_{\text{eff}}(3)/2 \cong 3.28942596$.

We first discuss the common features of both lattices and then analyse separately the two phase diagrams. The empty solid occurs in both phase diagrams. It undergoes a transition to a superfluid phase at finite $t$, where the critical value of $t$ increases monotonically with $|\mu|$. This value is solely determined by one-particle excitations [53]. For $\mu > 0$, there is a finite value of $t$ above which the phase becomes superfluid, with the exception of $\mu = 0$, where the phase is always superfluid. Recall that these features are also present in the phase diagram of the model with nearest-neighbour interactions.

### 5.2.1 Square lattice

By comparing Fig. 5 with the corresponding one for nearest-neighbour interactions on the square lattice, it is evident that the long-range interactions introduce several novel features. One striking feature is that the long-range interactions stabilises supersolid patterns. Most prominently, the $f = 1/2$ solid is surrounded by a $1/2$ supersolid. This phase has two sublattices arranged in a chequerboard pattern: one with sites with a minor occupation and the others with a major occupation.

Between the $f = 1/2$ and $f = 1/4$ solid, we also observe supersolids with a more complex sublattice structure. In general, it is hard to classify supersolid phases with a rich sublattice structure with the scheme introduced in Sec. 3. One reason for that is numerical noise on the output of the optimization routine and the necessity to introduce some tolerance to compare floating point numbers on a computer as equal. For larger unit cells, it might also occur that the modelling of a supersolid as a three- or four-sublattice supersolid is insufficient. This originates from slight deviations within those sublattices which results in a better energy minimization. Due to these effects, the automatised classification sometimes fails for certain supersolid states, for example at the fringes of the $f = 1/4$ solid. In most of these cases, one can assume that the respective supersolid phase is associated with the diagonal order of neighbouring solids.

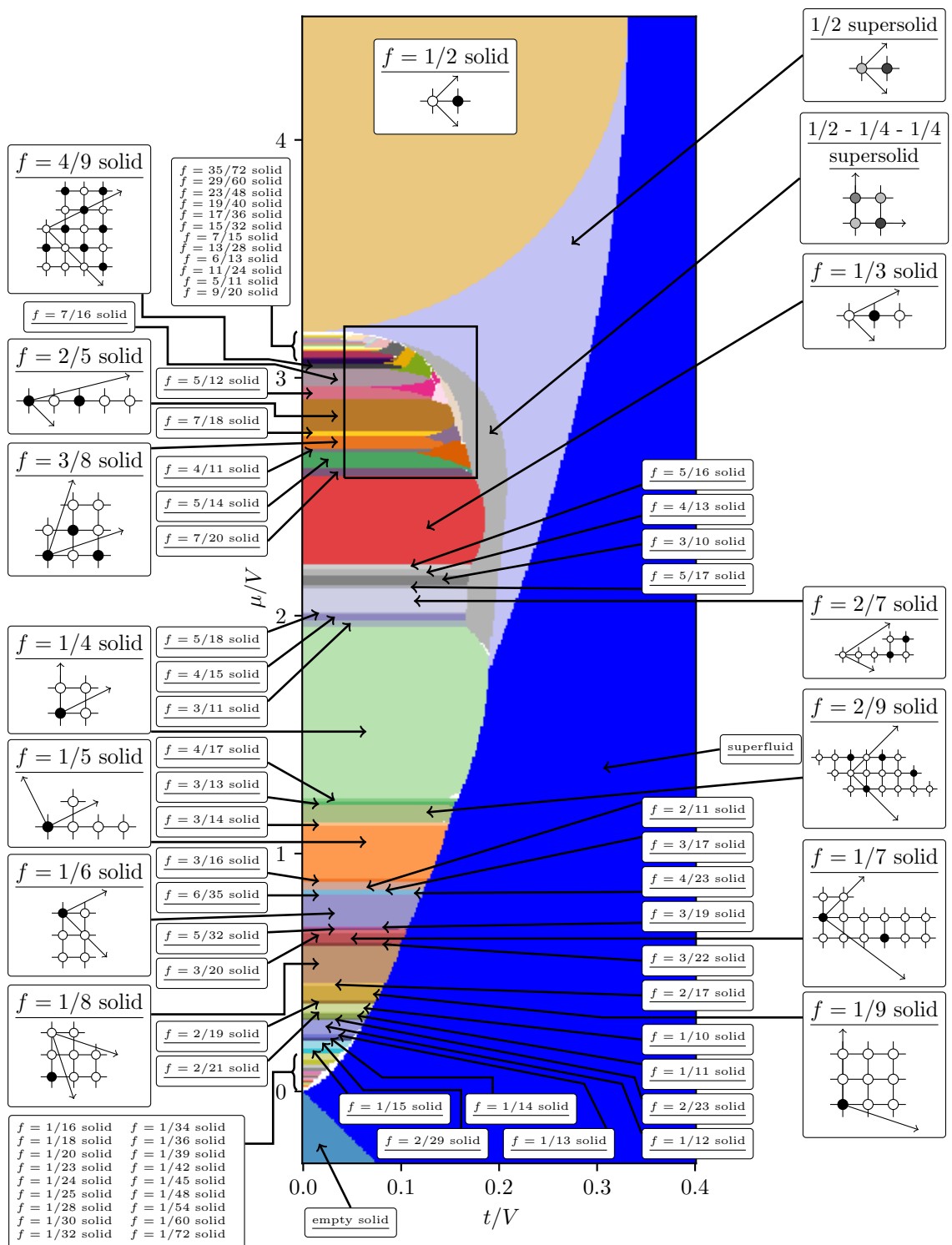

Figure 5: Quantum phase diagram in the $\mu/V - t/V$ plane for the square lattice. The colored regions indicate the different phases of the hardcore Bose-Hubbard model of Eq. (1) with dipolar interactions ($\alpha = 3$). The phases are determined using the classical spin mean-field approach. The colored phases are classified as described in Sec. 3. For the phases with a larger extent we also depict the unit cell with the corresponding translational vectors, if feasible. White regions correspond to the regime where our classification failed. The region in the black rectangle is zoomed in Fig. 6. The data $\{\theta_1, ..., \theta_N\}$, resulting from the numerical optimization procedure and used to create this figure, is reported in Ref. [64].

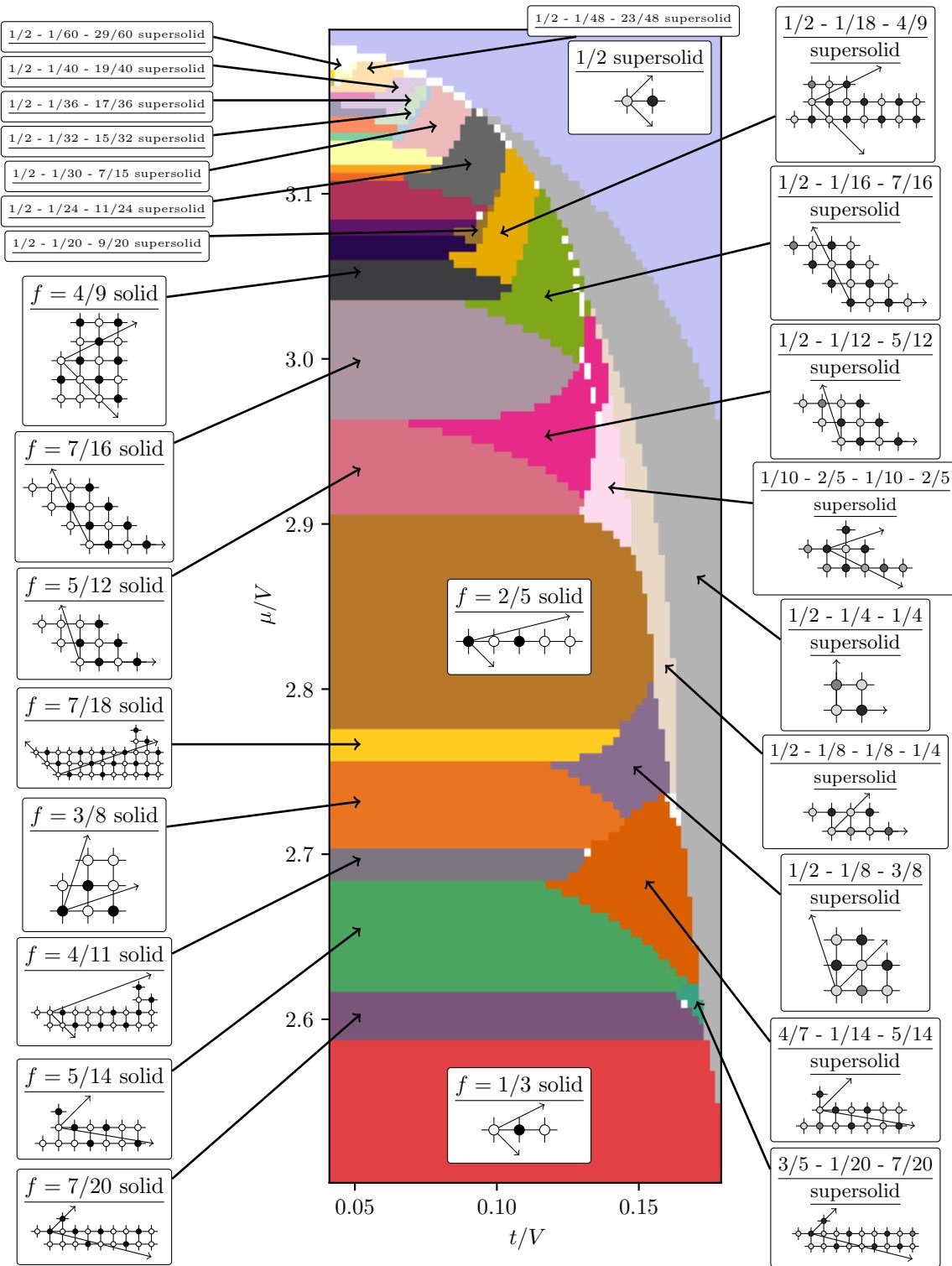

Figure 6: Zoom of the region inside the rectangle of Fig. 5.

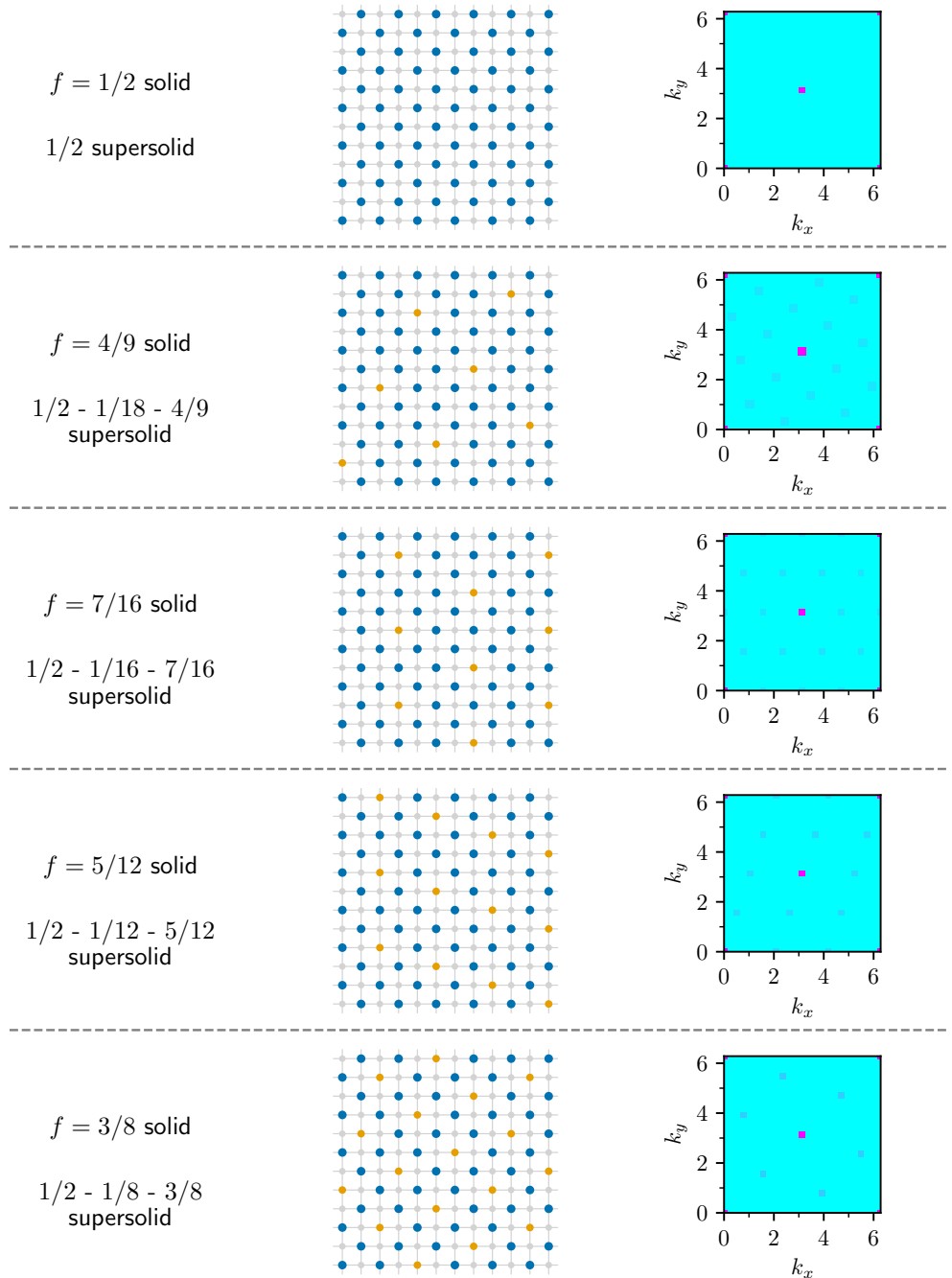

Figure 7: Detailed illustration of the structure of solid (supersolid) phases close to the chequerboard solid (supersolid), which can be regarded as defective chequerboard patterns. The central panels illustrate the density distribution in real space, the right panels the corresponding static structure factor (see Eq. (9)) in the $k_x - k_y$ plane and arbitrary units. In the central panel, for solid phases, the empty sites are gray, the filled sites are blue, while the orange circles indicate the empty sites which differentiate the structure from the chequerboard pattern and can be regarded as defects. For the supersolid, the colors indicate where the mean value of the density modulation is maximum (blue), minimum (gray), and at an in-between value (orange). On the right-hand side, the corresponding static structure factor (see Eq. (9)) in arbitrary units is depicted from $k_x, k_y \in [0, 2\pi]$. Pink indicates large amplitudes corresponding to the chequerboard pattern. The darker cyan indicated signatures associated with the defects.

Next, we focus on the plethora of solid and supersolid phases in the region within the highlighted rectangle in Fig. 5. Figure 6 zooms into this region. Notably, if there is a solid phase with an upwards-facing flank, there is a continuous transition to a supersolid phase with the same unit cell at the flank. These supersolids, that are directly related to the solid phases, are then surrounded by several more complex multi-sublattice supersolids. The latter, in turn, at some $t/V$ become a 1/2 supersolid.

We understand these transitions from the structural properties, starting from the consideration that the solid phases can be regarded as chequerboard patterns, but with some defects (see Fig. 7).

Take for example the $f = 5/12$ solid: Here, a full chequerboard pattern with $f = 1/2$ could be recovered by adding a particle to the second side of the uppermost row of the unit cell. In the corresponding supersolid, here the 1/2-1/12-5/12 solid, the mean local densities are redistributed in such a way that some density is transferred to the „defective" site. The tendency is to correct the defects of the chequerboard pattern. Note that we classify the supersolid phases using the structure factor (see Fig. 7), while the superfluid order parameter does not catch the differences.

We remark that there are not many instances of supersolid phases with several sublattices in the literature. Interestingly, such phases have been predicted in a frustrated quantum magnet and experimentally detected in $SrCu_2(BO_3)_2$ [95].

We conclude this section by comparing our results with the findings of Ref. [25]. In this paper the authors determined the phase diagram for the same Hamiltonian and geometry by means of quantum Monte Carlo, using finite square-shaped clusters with appropriately re-summed couplings similar to the procedure discussed in Sec. 3. The phase diagram reports $f = 1/2$, $f = 1/3$, and $f = 1/4$ solids as well as the 1/2 supersolid, and as far as it concerns these phases it is in qualitative agreement with our findings in Fig. 5. Direct comparison shows that the classical approximation, as expected, tends to overestimate the size of the solid phases along the axis $t/V$: The $f = 1/2$ solid lobe in Fig. 5 is about $10\%$ longer compared to Ref. [25] and the $f = 1/3$ and $f = 1/4$ solid lobes in Ref. [25] have about half the extent compared to Fig. 5. In more detail, the discrepancy between the classical approximation and the Monte Carlo results increases for solid phases with a larger unit cell. This could be traced back to the behaviour of the excitation gap, that becomes smaller for larger unit cells in the atomic limit $t = 0$ [96]. The classical approximation also overestimates the size of the supersolid phases: the $f = 1/2$ supersolid in [25] is a narrow stripe on the flanks of the $f = 1/2$ solid lobe, while in Fig. 5 it stretches down along the $\mu$ axis till the $f = 1/4$ solid. The limit $t/V \to 0$ is not commented in Ref. [25], presumably because of the breakdown of the quantum Monte Carlo approach due to the lack of off-diagonal operators in the simulation. This is the limit where we expect that the algorithm of Sec. 3 becomes exact and can provide a viable complement to more elaborate approaches.

### 5.2.2 Honeycomb lattice

Figure 8 displays the phase diagram for the honeycomb lattice, as we determine it using our method. Its overall appearance is similar to the phase diagram of the square lattice in Fig. 5. We attribute this similarity to the fact that both lattices are bipartite. Similar to the square lattice, also here we find a 1/2 supersolid phase enveloping the solid phases between the $f = 1/2$ and the $f = 1/4$ solid. In contrast to the square lattice, we do not observe large variety of supersolid phases between the solid phases and the 1/2 supersolid.

In general, on the honeycomb lattice it is especially hard to automatically identify supersolid phases, even harder than for the square or the triangular lattice. This could

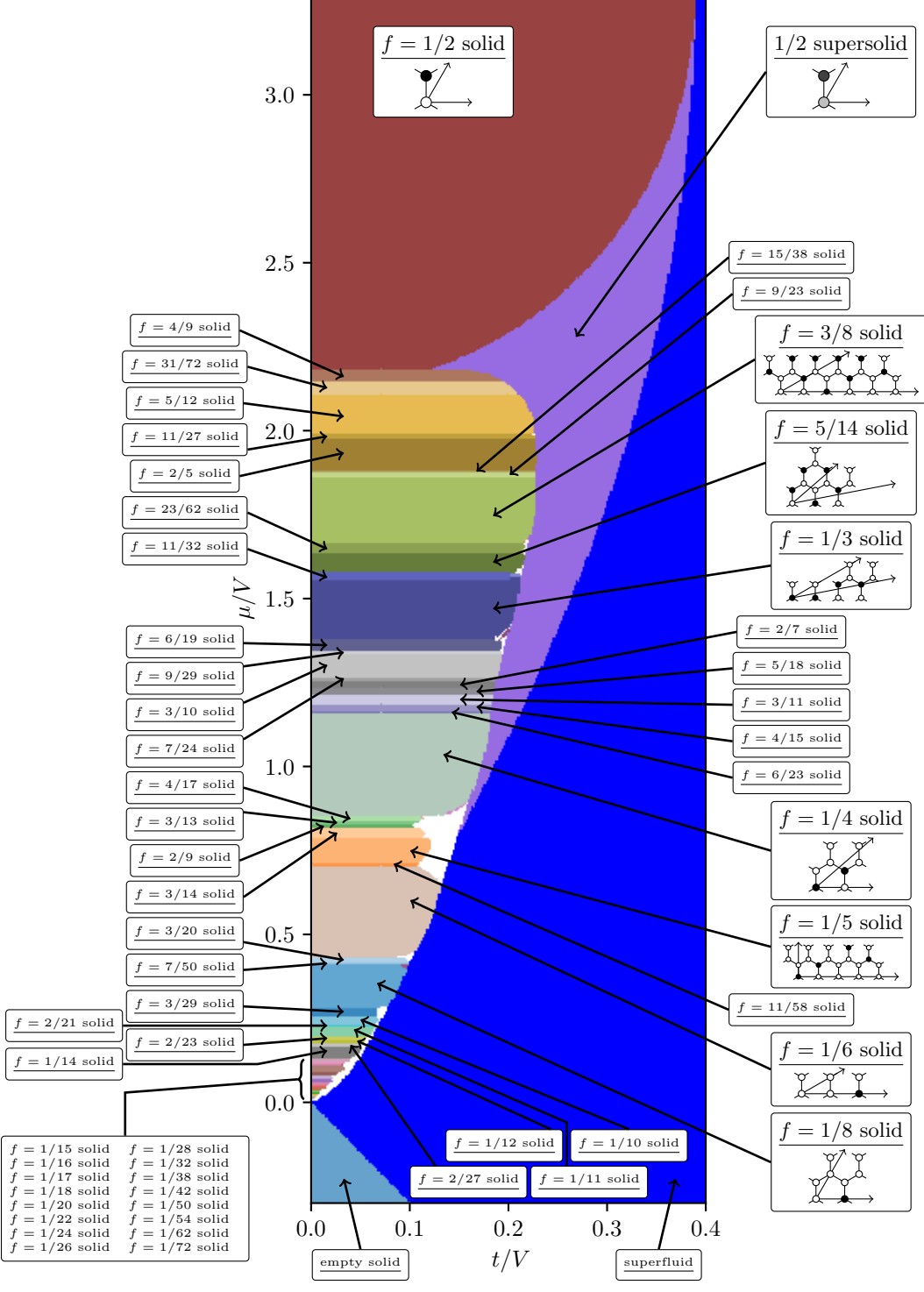

Figure 8: Quantum phase diagram in the $\mu/V - t/V$ plane for the honeycomb lattice. The colored regions indicate the different phases of the hardcore Bose-Hubbard model of Eq. (1) with dipolar interactions ($\alpha = 3$). The phases are determined using the classical spin mean-field approach. The colored phases are classified as described in Sec. 3. For the phases with a larger extent we also depict the unit cell with the corresponding translational vectors, if feasible. White regions correspond to the regime where our classification failed. The data $\{\theta_1, ..., \theta_N\}$, resulting from the numerical optimization procedure used to create this figure, is reported in Ref. [64].

be due to the overall larger number of sites per unit cells of the honeycomb lattice. Also, the minimization routine is numerically more expensive for the honeycomb lattice since the number of sites is doubled in comparison to the triangular lattice. We expect that the phases on the tips of the solids are supersolid with a complex sublattice structure. Regarding the not-identified points in the phase diagrams, the unit cells that are reported have a very large extent and seem to be bound by the size of unit cells that is used in our scheme. This poses a natural limitation to the method. To get a feeling for the computational effort of the numerical optimisation procedure on the honeycomb lattice: It required several 10000 CPUh. Therefore, an investigation of larger unit cells is impractical. As a consequence, we cannot make any further statements about the not-identified areas except that they display supersolid behaviour and that they cannot be easily classified according to the simple metrics implemented in this work.

# 6 Quantum phase diagrams of long-range interacting bosons: triangular lattice

In this section, we analyse the phase diagram for the triangular lattice and repulsive dipolar density-density interactions. Figure 9 displays the fractional solids as a function of $\mu$ in the atomic limit $t = 0$. The phase diagram is drawn till the particle-hole symmetry line, which is here located at $\mu/V = z_{\text{eff}}(3)/2 \cong 5.517087866$. The fractional solids form a staircase, where the solid at $f = 1/3$ has the largest plateau. Compared to the other geometries, see Fig. 3, this is a prominent distinctive feature. A more detailed analysis shows that the interplay between long-range interactions and the triangular lattice geometry tends to stabilise patterns with a hexagonal unit cell, e. g., the $f = 1/3$ solid, the $f = 1/4$ solid, the $f = 1/7$ solid or the $f = 1/9$ solid, see the lower part of Fig. 9. We attribute this effect to the optimal equidistant distribution of particles in a hexagonal structure.

The phase diagram in the $\mu/V - t/V$ plane is shown in Fig. 10. We first observe the empty solid phase at $\mu < 0$, which is a common feature with the phase diagrams of the corresponding model with nearest-neighbour interactions. A further common feature is that the phase is superfluid at sufficiently large values of $t$ depending on $\mu$. Also for the triangular lattice, at $\mu = 0$ (and at the corresponding symmetric value) we observe no phase transitions along the $t$ axis: The phase is always superfluid.

In order to identify the features due to the long-range interactions we now compare the phase diagram of Fig. 10 with the one obtained for nearest-neighbour interactions in Fig. 2. Both phase diagrams exhibit the $f = 1/3$ solid and supersolid phases. In addition, the long-range interactions give rise to a staircase of gapped solid phases [34]. The $f = 1/2$ solid phase, in particular, is found at the particle-hole symmetry line and has the features of a stripe pattern (see Fig. 9). Interestingly, in this parameter regime the model can be mapped to the antiferromagnetic long-range Ising model, where stripe patterns have recently been conjectured [38–41] and numerically confirmed [34]. By inspecting the behaviour of the $f = 1/2$ solid in Fig. 10 as a function of $t$, we observe a transition to a supersolid state with three sublattice structure. This supersolid phase is absent in the nearest-neighbour model and has a phase transition to a superfluid phase at sufficiently large values of the tunnelling amplitude.

Both nearest-neighbour and dipolar model predict a supersolid phase at $f = 1/3$. In the long-range model it now envelopes all phases above the chemical potential of the $f = 1/3$ solid and also appears below the $f = 1/3$ lobe till the tip of the $f = 1/4$ solid. Further, we find a 1/4 supersolid on the lower flank of the $f = 1/4$ solid.

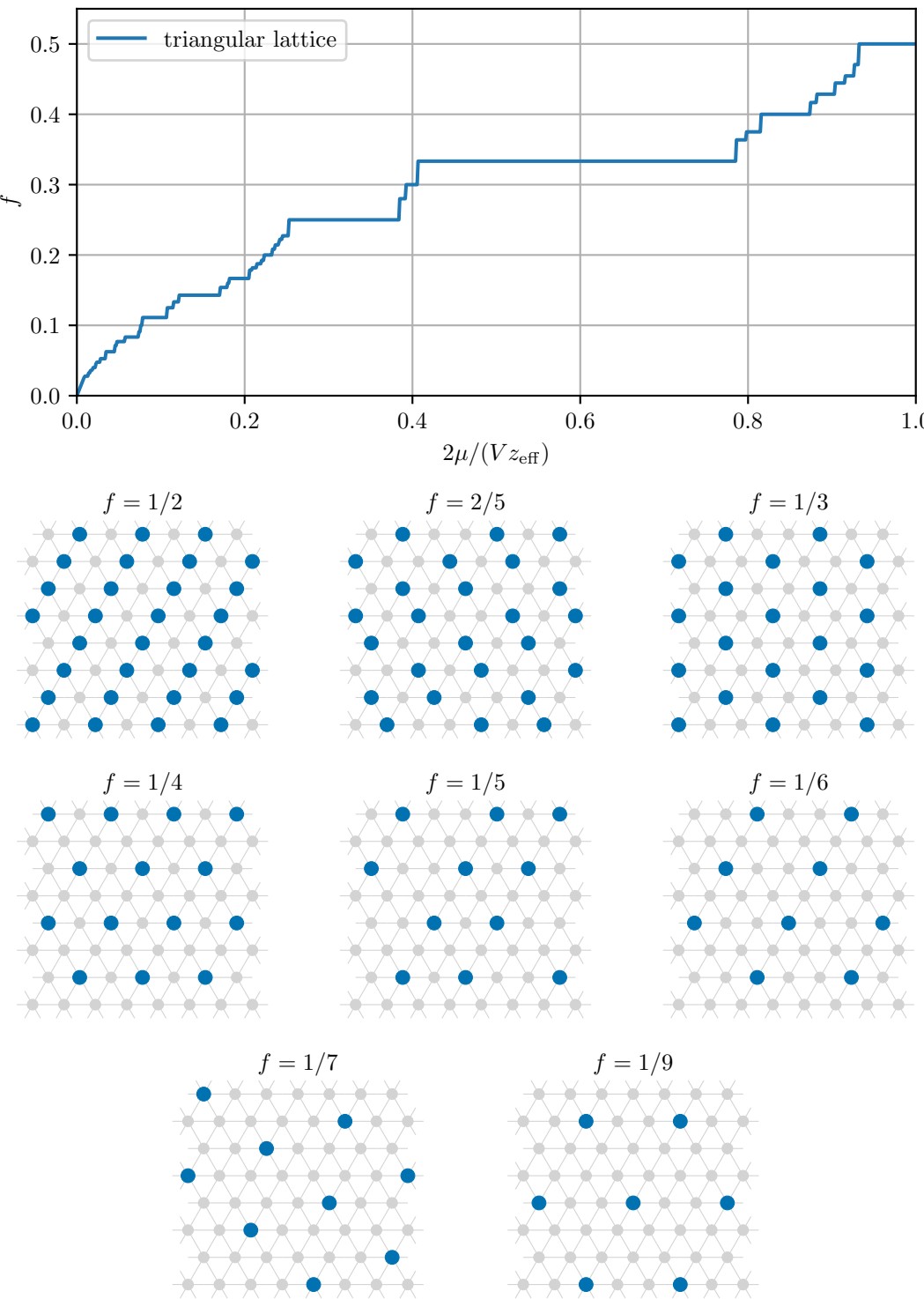

Figure 9: Devil's staircase of solid phases with filling $f$ of the hardcore Bose-Hubbard model with repulsive dipolar long-range density-density interactions at $t/V = 0$ on the triangular lattice. The staircase in $\mu/V$ of the triangular lattice is normalised to the particle-hole symmetry line. In the panels below the plot, we depict some some phases of the staircase. Blue circles indicate occupied sites, light grey circles the empty sites.

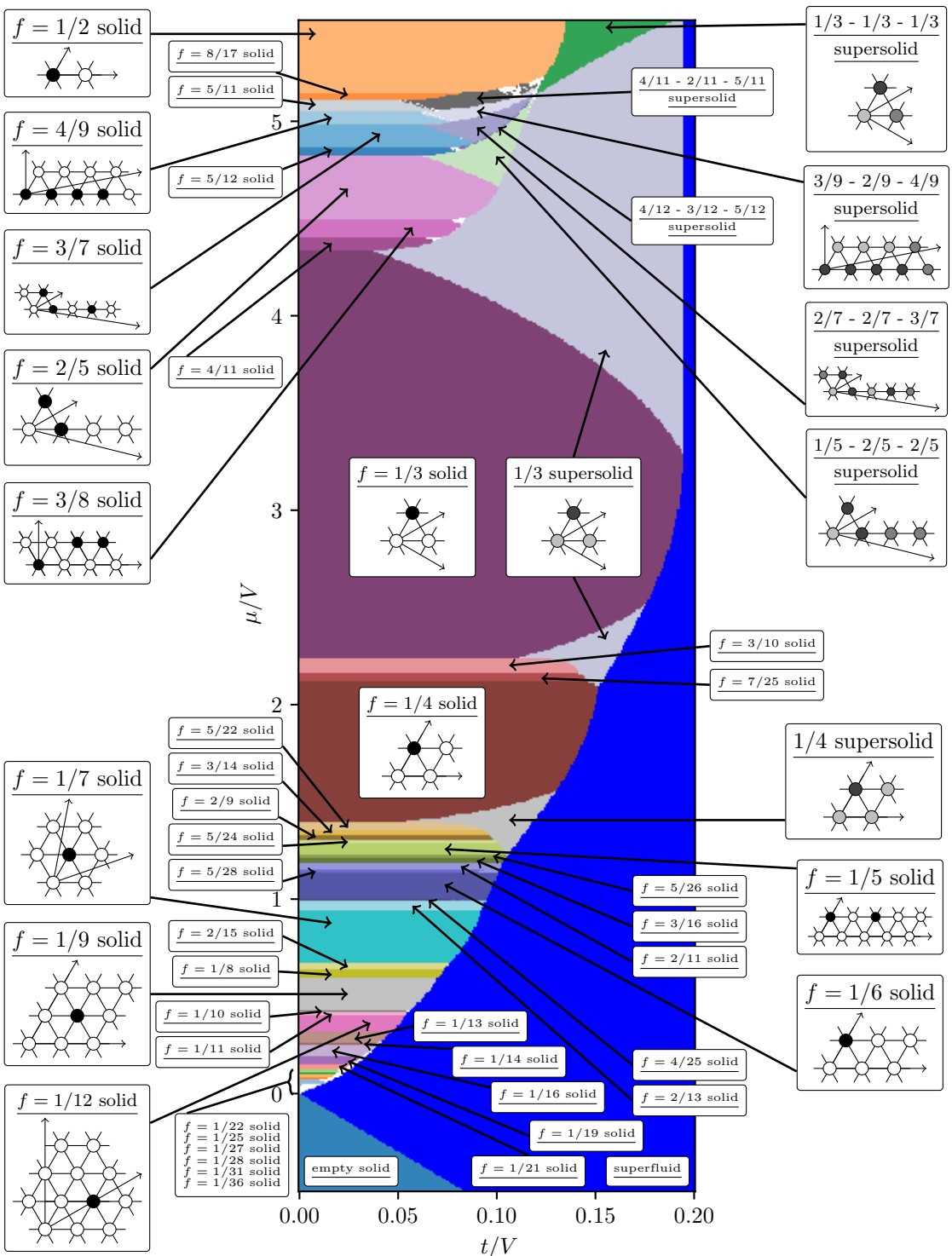

Figure 10: Quantum phase diagram in the $\mu/V - t/V$ plane for the triangular lattice. The colored regions indicate the different phases of the hardcore Bose-Hubbard model of Eq. (1) with dipolar interactions ($\alpha = 3$). The phases are determined using the classical spin mean-field approach. The colored phases are classified as described in Sec. 3. For the phases with a larger extent we also depict the unit cell with the corresponding translational vectors, if feasible. White regions correspond to the regime where our classification failed. The data $\{\theta_1, ..., \theta_N\}$, resulting from the numerical optimization procedure used to create this figure, is reported in Ref. [64].

A detailed analysis of the region between the $f = 1/2$ and $f = 1/3$ solid shows, that at the upper flank of the tips of the solid phases there occur supersolid states with the same unit cell.

We now connect our results with a quantum Monte Carlo study by L. Pollet et al. [36], investigating the ground-state phase diagram around the $f = 1/3$ lobe by means of the worm algorithm [97,98]. In this study, the authors calculated the structure factor associated with the $f = 1/3$ solid, the mean occupation, and the superfluid stiffness to characterise the differences between the $f = 1/3$ solid, $1/3$ supersolid, and superfluid (analogously to Tab. 1). For the $f = 1/3$ solid Ref. [36] reports a maximal extent of $t/V \approx 0.13$ while the classical mean-field approximation gives a larger value of $t/V \approx 0.19$. This is in line with the expectation that quantum fluctuations reduce the stability of solid phases and can be seen as well for the nearest-neighbour model (see Fig. 2). Similar to the nearest-neighbour model Ref. [36] reports the $1/3$ supersolid phase enveloping the $f = 1/3$ solid at its upper flank. The extent of the $1/3$ supersolid is reported to have a dip until $t/V \approx 0.065$ for $\mu/V \approx 5.3$ until its extent grows to $t/V \approx 0.08$ at the particle-hole symmetry line. Note, Ref. [36] reports a mean occupation of $1/2$ around the particle hole symmetry line. Further, Ref. [36] reports a breakdown of superfluidity and the $f = 1/3$ order parameter for small $t/V$ close to the particle-hole symmetry line. These reports are in qualitative agreement with the mean-field phase diagram, since the $f = 1/2$ stripe order, as well as some of the solid orders between the $f = 1/3$ and $f = 1/2$ cannot be measured with the ordering vector of the $f = 1/3$ solid. Also, the $1/3$ - $1/3$ - $1/3$ supersolid cannot be distinguished from the $1/3$ supersolid using the set of observables described in Ref. [36]. From the statements in Ref. [36] we can also conclude that we overestimate the extent of the $f = 1/2$ solid which should be around $t/V \approx 1/30$. The qualitative agreement with Ref. [36] shows that our approach is a reliable ansatz for a first analysis of phase diagrams with long-range interactions, preceding an accurate numerical study, and generally shows that our calculations allow to get insight into the emerging phases and their dependence on the tunable parameters. In order to pinpoint the precise nature of the occurring solid and supersolid phases it is beneficial to know in advance the relevant ordering vectors (see also the discussion in Sec. 5.2.1). Therefore, one shall fist check occurring orders and their unit cells with the method discussed in this work. Besides the existing quantum Monte Carlo results in Ref. [36], a follow-up study using quantum Monte Carlo techniques would be desirable to shed light, for instance, on the stability of the emergent $1/3 - 1/3 - 1/3$ supersolid at the particle-hole symmetry line.

# 7 Conclusions

In this work, we calculated ground-state quantum phase diagrams of the hardcore Bose-Hubbard model with repulsive dipolar density-density interactions on the square, honeycomb, and triangular lattice. We extended the systematic approach presented in Ref. [34] to finite hopping amplitudes and perform the classical spin approach without a truncation of the long-range interaction. As we do not truncate the long-range interaction and consider all possible unit cells up to a given extent, we have access to the devil's staircase of solid phases arising from the long-range interaction in the limit of small hoppings [34]. The method in use is exact in the limit of $t/V = 0$ and is only limited by the size of the considered unit cells. Further, the classical spin approach provides access to a full quantum phase diagram in $t/V$ and $\mu/V$. We found a plethora of possible supersolid phases at some intermediate $t/V$, which are separated from a superfluid phase at sufficiently large $t/V$. A remarkable finding is the occurrence of supersolid phases in the classical spin approach

which have more than two sublattices at different mean densities.

Our predictions are relevant for experimental platforms realising ultracold Bose gases with dipolar atoms [99–101] or ground-state polar molecules [102–104]. In these systems, Bose-Hubbard models have been realised [104,105]. The onsite interaction can be tuned to zero by means of a Feshbach resonance, and the tunneling rate can be changed by varying the depth of the lattice [10]. The chemical potential, finally, can be modified by external fields shifting the onsite energy [10]. Long-range diagonal order is typically probed by Bragg spectroscopy [10]. Off-diagonal order is accessed using time-of-flight measurements [10].

The results presented in this work for dipolar density-density interactions shed light on the interplay of frustration and long-range interactions, showing that they lead to a plethora of solid and supersolid phases, otherwise absent in the corresponding nearest-neighbor model. Clearly, the classical spin approach does significantly underestimate the effect of quantum fluctuations. When comparing to quantum Monte Carlo studies [25,27,36,62,63], the extent in $t/V$ of solids with larger unit cells and especially superfluids is significantly reduced. Nevertheless, the phase diagrams presented in this work provide an important puzzle piece for further investigations on this problem, as we list all the possible occurring phases. In our study, we have demonstrated that it is important to carefully choose observables for further numerical studies since many of the distinct complex supersolid phases (see Fig. 6 and Fig. 7) have similar correlations. The precise extent and stability of the presented phases need to be tied down by subsequent large-scale numerical studies capturing the quantum fluctuations. Especially for the triangular lattice the existing quantum Monte Carlo study calculated only three simple observables [36]. These subsequent studies can then use the results of this work in order to adjust the choice of their unit cells and observables in order not to miss any phase.

In the context of the recent experimental findings by Su et al. [33], the method in Sec. 3 and the phase diagrams in Fig. 5 and Fig. 6 for the square lattice pose a powerful framework for studying more complex dipolar solids. In an experiment, one could first focus on the $f = 1/3$ and $f = 1/4$ solids, as these solid phases are expected to be stable against moderate quantum fluctuations [25].

Our analysis can be simply extended to configurations with tilted angles, as in the experimental study of Ref. [33]. Our framework allows us to complement existing studies [29, 30, 106] thereby unveiling the interplay between long-range interactions, their anisotropy, and frustration. Our model can be further extended to include the effect of long-range correlated tunnelling., which give rise to staggered superfluid phases for nearest-neighbour interactions [107].

With the recent development of efficient algorithms [70–73] for the calculation of the resummed couplings one shall be able to treat a larger number of unit-cells with larger sizes. Then the global optimizer on each unit cell will become the limiting factor of the calculations. Our present aim is to extend our unit-cell-based approach to better account for quantum fluctuations.

Therefore, future works shall analyse whether and how the resummed coupling unit cell approach can be integrated with more sophisticated mean-field calculations [61, 90, 108, 109] and with other numerical methods such as exact diagonalisation [110], quantum Monte Carlo schemes [25, 27, 62, 63, 110, 111] or continuous similarity transformations [93, 112]. Another interesting research direction would be the study of two-dimensional devil's staircases without particle-hole symmetry, by transferring the ideas of Ref. [113] for one-dimensional systems.

# 8 Acknowledgements

The authors thank Raphaël Menu and Tom Schmit for fruitful discussions. JAK thanks Andreas Buchheit, Jonathan Busse, and Ruben Gutendorf for fruitful discussions on the Epstein $\zeta$-function. The authors greatfully acknowledge the scientific support and HPC resources provided by the Erlangen National High Performance Computing Center (NHR@FAU) of the Friedrich-Alexander-Universität Erlangen-Nürnberg (FAU).

**Funding information**  This work was funded by the Deutsche Forschungsgemeinschaft (DFG, German Research Foundation) - Project-ID 429529648 - TRR 306 QuCoLiMa (Quantum Cooperativity of Light and Matter) and in part by the National Science Foundation under Grants No. NSF PHY-1748958 and PHY-2309135. KPS and JAK acknowledge the support by the Munich Quantum Valley, which is supported by the Bavarian state government with funds from the Hightech Agenda Bayern Plus. The hardware of NHR@FAU is funded by the German Research Foundation DFG.

**Data availability**  The unit cells with the appropriately resummed couplings used for the classical spin mean-field calculations in this work are available in Ref. [64]. The results of the optimisation procedure used to draw the phase diagrams published in this work are also available in Ref. [64].

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
