# Peer review of "Quantum phases of hardcore bosons with repulsive dipolar density-density interactions on two-dimensional lattices"

_SciPost Physics_

## Round 1 · Referee Report · Anonymous (Referee 1) · 2024-2-2

Strengths

1) A detailed mean-field analysis of dipolar bosons particularly in a triangular lattice where QMC data does not exist.

Weaknesses

1) The paper seems to be a possible and somewhat expected application of the formalism developed in Ref 34 of the work (by the same authors).

2) The mean field phase diagrams of dipolar bosons has been worked out by several authors in the phase predicting existence of supersolid phase and devil staircase structures. See for example Phys. Rev. A 83, 013627 (2011), New J. Phys. 17 123014 (2015) or Europhys. Lett. 87 36002 (2009). In fact there are many other similar results in the literature.

Report

I believe, based on the weaknesses mentioned, the paper in its present form is suitable for publication in Scipost Phys. Core. Whereas the analysis is detaiked and it deserves publication in some form, I do not see sufficiently new results which allows consideration in Scipost.
  • validity: good
  • significance: ok
  • originality: ok
  • clarity: good
  • formatting: good
  • grammar: excellent

Author:  Jan Alexander Koziol  on 2024-07-12  [id 4617]

(in reply to Report 1 on 2024-02-02)

We thank the referee for the report. We have modified the text accordingly in order to address the referees remarks and criticisms.

We acknowledge that the referee recommends the publication of our work in SciPost Phys. Core. However, we disagree and believe that this assessment is mostly due to the presentation of our results, which did not sufficiently clarified their novelty. For this purpose, we have revised introduction and the text in order to emphasize the results we obtained.

As far as it concerns the relation to SciPost Phys. 14, 136 (2023): we now include quantum quantum fluctuations and study the interplay of quantum fluctuations, long-range interactions, and frustration. The novelty of the present work lies thus not only on the methodology, but also on the results we obtain.

We summarize our results:

We identify a number of features that are not captured by the nearest-neighbor truncation and unveil the effect of the interplay between the long-range dipolar interactions with the lattice geometry.

In the limit of vanishing tunnelling, we show that the ground state is a devil's staircase of solid phases, which can be identified up to an arbitrary precision. This precision, in fact, is only limited by the size of the considered unit cells and the optimization scheme we employ. To the best of our knowledge, we are not aware of studies mapping out quantitatively the complete devil's staircase for two-dimensional systems.

For finite tunnelling, we find solid and supersolid phases, some of which have been reported by numerical studies based on advanced quantum Monte Carlo simulations. Differing from these works, we can avoid the limitation imposed by the constraints on the unit cell and thereby unveil a plethora of solid and supersolid phases which have not been reported before. We characterize the corresponding phases, and unveil a very structured phase diagram. Besides the fundamental interest, our results will be a guide for experimentalists working in the field of dipolar gases as well as frustrated magnets.

Finally, our results thus also provide an important benchmark and guidance for numerical programs, identifying the relevant unit cells, simulation geometries, and observables.

Let us finally state that we fully agree with the referee that "The mean field phase diagrams of dipolar bosons has been worked out by several authors in the phase predicting existence of supersolid phase and devil staircase structures". Nevertheless, the majority of these works truncated the dipolar interactions to the nearest-neighbour, as is the case for some of the ones the referee mentions. Indeed, a result of our paper is in line with advanced numerical studies, and show that the nearest-neighbor approximation for dipolar interactions misses to identify multi-lattice solid and supersolid phases, that our resummation approach instead allows to capture.

We have added these references to the conclusions, in the paragraph where we discuss realizations with tilted dipoles.

Let us further comment on the references mentioned by the referee:

  • Thieleman et al., PRA 83, 013627 (2011): The authors focus on nearest-neighbor interactions and a complex hopping as well as cold atoms in a staggered flux. In our work, we study a different model because we have no staggered flux. At the same time, we concentrate on the effects of long-range interactions, thus no truncation in the power-law is performed.

  • Zhang et al., New J. Phys. 17 123014 (2015: The authors apply QMC for dipolar hardcore bosons on the square lattice, but restrict to half filling. In our work, we investigate the whole phase diagram at any filling for several 2d lattice geometries.

  • Isakov et al., Europhys. Lett. 87 36002 (2009): The authors apply variational QMC and Schwinger-boson mean-field theory for half-filling and anisotropic nearest-neighbor interactions giving rise to a staircase of phases. Again, we focus on the effects of the untruncated long-range interactions.

---

## Round 1 · Referee Report · Anonymous (Referee 2) · 2024-2-26

Strengths

1) The manuscript is clearly written. A brief review of the method used is provided. All relevant physics is neatly summarized in the first sections of the manuscript.

2) Citations are carefully chosen and very relevant to the topic.

3) Results are for the most part clearly described.

4) The results of this work are interesting and relevant to current experiments with ultracold gases.

5) The manuscript provides novel in depth analysis of the strongly interacting regime (where the method presented in most accurate). A plethora of solid phases are presented which had not been observed before with approximation-free methods.

Weaknesses

1) On the one hand, significant emphasis is given to QUANTUM phases, such as supersolid phases, in the introductory part of the manuscript. On the other hand, most of the description of results is dedicated to solid phases. Only supersolid phases at 1/2 (for bipartite) and 1/3,1/4 (for non-bipartite) are marked in the phase diagrams. While the 1/4 supersolid in triangular lattice is new, the others were already known. It would be better to further describe novel supersolids such as the ones mentioned for square lattice (for honeycomb lattice limitations of the method are discussed). For example, what are supersolids "with more complex sub lattice structure"? Maybe a picture of density patterns of this supersolid would help. If it is already there, then it is not clear that it does refer to a supersolid phase. Moreover, the authors should comment about how their results compare with QMC study of hard-core purely repulsive dipolar bosons in triangular lattice (PRL 104, 125302 (2010)).

Report

Overall, this is an interesting, well-written manuscript which presents results on many-body hamiltonians relevant to current experimental efforts in ultracold gases. It meets the criteria for publication of SciPost and I therefore recommend it for publication once the requested changes are addressed.

Requested changes

1) Do authors have an explanation on why their method does not compare as well with approximation-free numerical methods in the case of non-bipartite triangular lattice?

2) Why is the method so much more accurate at the Heisenberg point?

3) Can the authors address the point mentioned in the Weaknesses Section of this report?

  • validity: high
  • significance: high
  • originality: high
  • clarity: high
  • formatting: excellent
  • grammar: excellent

Author:  Jan Alexander Koziol  on 2024-07-12  [id 4618]

(in reply to Report 2 on 2024-02-26)

We thank the referee for the report. We have modified the text accordingly in order to address the referees remarks and criticisms.

Below we address the points raised by the referee.

i) Regarding the nearest-neighbour case: the literature states that quantum fluctuations have a larger impact on systems with geometrical frustration. The model on the triangular lattice is subject to geometrical frustration, therefore a larger impact of quantum fluctuation on the phase diagram is expected. We have added a clarification of this point in Sec. 5.

ii) The classical approximation captures the transition at the Heisenberg point correctly since the transition is driven by the change in symmetry of the Hamiltonian from an easy-axis to a rotationally invariant interaction. This change in symmetry is present in the classical spin picture, as well as, the full quantum mechanical problem for the same parameter value $t/V=1/2$. We have added a clarification of this point in Sec. 5.

iii) We have strengthened the emphasis on the quantum phases in the results section. We realised that we used the misleading name "order" for "supersolid" phases in Figs. 5, 6, 10. We correspondingly modified it in the figure captions and in the text. Further, we added text passages discussing the different supersolid phases for the square and the triangular lattice.

In Fig. 7, we have added real and momentum space depictions of the complex supersolid phases on the square lattice. We have added a picture how to understand the supersolid phases on the square lattice in terms of defective checkerboard patterns.

Regarding the triangular lattice, we are grateful for the reference to the quantum Monte Carlo study PRL 104, 125302 (2010). We have added a discussion of the reference in comparison to our results in Sec. 7. In the conclusion we highlighted the application of the discussed method to determine relevant observables and unit cells for further numerical studies.

---

## Round 2 · Referee Report · Anonymous (Referee 3) · 2024-7-16

Strengths

  1. The paper is well-organized, with clear sections.
  2. The study addresses contemporary topics in quantum simulation and condensed matter physics, particularly with relevance to experimental realizations with ultracold dipolar atoms.
  3. The use of a classical spin mean-field approach without truncating the long-range interactions is a significant strength, offering detailed insights into the quantum phases.
  4. The paper provides a thorough investigation of the ground-state quantum phase diagram for hardcore bosons with repulsive dipolar potentials on various two-dimensional lattices.
  5. The comparison with known quantum phase diagrams and quantum Monte Carlo simulations adds robustness to the findings.

Weaknesses

  1. While the paper discusses experimental relevance, more detailed guidance on how these theoretical predictions can be implemented experimentally would be beneficial.
  2. The precision of the results is constrained by the size of the unit cells and the optimization scheme, which may limit the generalizability of the findings.

Report

The authors utilize a classical spin mean-field approach to account for long-range dipolar interactions without truncation, providing a detailed analysis of the ground states on square, honeycomb, and triangular lattices.

The paper's strengths lie in its comprehensive approach and innovative methodology. The authors' comparison with existing quantum phase diagrams and Monte Carlo simulations adds credibility to their results. The clear and structured presentation enhances readability, guiding the reader through complex analyses effectively.

However, the paper has some weaknesses which I have provided under the "Requested changes".

Requested changes

  1. Address Mean-Field Limitations: If possible, provide a more detailed discussion on the limitations of the classical mean-field approach, particularly concerning quantum fluctuations.

  2. Detail Experimental Implementation: If possible, offer more specific guidance on how the theoretical predictions can be realized experimentally, possibly with examples or proposed setups.

  3. If possible, discuss the implications of the computational constraints in more detail and suggest potential ways to overcome these limitations in future research.

Recommendation

Ask for minor revision

  • validity: top
  • significance: top
  • originality: top
  • clarity: top
  • formatting: perfect
  • grammar: perfect

Author:  Jan Alexander Koziol  on 2024-09-12  [id 4769]

(in reply to Report 1 on 2024-07-16)

Response: Anonymous Report 1 on 2024-7-16 (Contributed Report)

We thank the referee for the report.

We have modified the text accordingly to address the requested changes by the referee.

Regarding point 1: We thank the referee for raising this point. We address the limitations of the classical mean-field approach in Sec. 3.4 of the manuscript. To the already existing discussion, we added an emphasis that in our work we compare our results to quantum Monte Carlo results for the square and triangular lattice to gauge the effect of quantum fluctuations. Furthermore, we added general statements that an increased coordination number, system dimension, or spin quantum number will decrease the strength of the quantum fluctuations.

Regarding point 2: We thank the referee for this point. We added a paragraph to the conclusion of the manuscript, where we outline how ultracold dipolar atoms or molecules can be used to realize our model. We summarize how to experimentally tune model parameters and how to experimentally probe diagonal and off-diagonal long-range order in the model.

Regarding point 3: We thank the referee for this point. Indeed, we have identified multiple ways to potentially improve our numerical analysis, which we have added on several occasions in the manuscript. First, we became aware of a novel algorithm to calculate the resummed couplings, and, therefore, more unit cells with more lattice sites will be available. Second, one can try to adjust the choice of the global optimizer for the classical energy functions, which is the current bottleneck of the calculations. Third and most importantly, our focus will shift to the transfer of the unit-cell-based approach to numerical methods that better account for quantum fluctuations.

---

## Round 2 · Referee Report · Anonymous (Referee 2) · 2024-8-26

Report

The Authors have significantly revised the original manuscript and have answered the questions and comments in a satisfactory manner. I believe the results reported are interesting and based on the journal acceptance criteria, I recommend the manuscript for publication.

Recommendation

Publish (meets expectations and criteria for this Journal)

  • validity: -
  • significance: -
  • originality: -
  • clarity: -
  • formatting: -
  • grammar: -

Author:  Jan Alexander Koziol  on 2024-09-12  [id 4770]

(in reply to Report 2 on 2024-08-26)

Response: Anonymous Report 2 on 2024-8-26 (Contributed Report)

We thank the referee for the positive feedback to our work and the recommendation for publication.

---

## Round 2 · Author Response

Warnings issued while processing user-supplied markup:

  • Inconsistency: Markdown and reStructuredText syntaxes are mixed. Markdown will be used.
    Add "#coerce:reST" or "#coerce:plain" as the first line of your text to force reStructuredText or no markup.
    You may also contact the helpdesk if the formatting is incorrect and you are unable to edit your text.

Response: Anonymous Report 1 on 2024-2-2 (Invited Report)

We thank the referee for the report. We have modified the text accordingly in order to address the referees remarks and criticisms.

We acknowledge that the referee recommends the publication of our work in SciPost Phys. Core. However, we disagree and believe that this assessment is mostly due to the presentation of our results, which did not sufficiently clarified their novelty. For this purpose, we have revised introduction and the text in order to emphasize the results we obtained.

As far as it concerns the relation to SciPost Phys. 14, 136 (2023): we now include quantum quantum fluctuations and study the interplay of quantum fluctuations, long-range interactions, and frustration. The novelty of the present work lies thus not only on the methodology, but also on the results we obtain.

We summarize our results:

We identify a number of features that are not captured by the nearest-neighbor truncation and unveil the effect of the interplay between the long-range dipolar interactions with the lattice geometry.

In the limit of vanishing tunnelling, we show that the ground state is a devil's staircase of solid phases, which can be identified up to an arbitrary precision. This precision, in fact, is only limited by the size of the considered unit cells and the optimization scheme we employ. To the best of our knowledge, we are not aware of studies mapping out quantitatively the complete devil's staircase for two-dimensional systems.

For finite tunnelling, we find solid and supersolid phases, some of which have been reported by numerical studies based on advanced quantum Monte Carlo simulations. Differing from these works, we can avoid the limitation imposed by the constraints on the unit cell and thereby unveil a plethora of solid and supersolid phases which have not been reported before. We characterize the corresponding phases, and unveil a very structured phase diagram. Besides the fundamental interest, our results will be a guide for experimentalists working in the field of dipolar gases as well as frustrated magnets.

Finally, our results thus also provide an important benchmark and guidance for numerical programs, identifying the relevant unit cells, simulation geometries, and observables.

Let us finally state that we fully agree with the referee that "The mean field phase diagrams of dipolar bosons has been worked out by several authors in the phase predicting existence of supersolid phase and devil staircase structures". Nevertheless, the majority of these works truncated the dipolar interactions to the nearest-neighbour, as is the case for some of the ones the referee mentions. Indeed, a result of our paper is in line with advanced numerical studies, and show that the nearest-neighbor approximation for dipolar interactions misses to identify multi-lattice solid and supersolid phases, that our resummation approach instead allows to capture.

We have added these references to the conclusions, in the paragraph where we discuss realizations with tilted dipoles.

Let us further comment on the references mentioned by the referee:

  • Thieleman et al., PRA 83, 013627 (2011): The authors focus on nearest-neighbor interactions and a complex hopping as well as cold atoms in a staggered flux. In our work, we study a different model because we have no staggered flux. At the same time, we concentrate on the effects of long-range interactions, thus no truncation in the power-law is performed.

  • Zhang et al., New J. Phys. 17 123014 (2015: The authors apply QMC for dipolar hardcore bosons on the square lattice, but restrict to half filling. In our work, we investigate the whole phase diagram at any filling for several 2d lattice geometries.

-Isakov et al., Europhys. Lett. 87 36002 (2009): The authors apply variational QMC and Schwinger-boson mean-field theory for half-filling and anisotropic nearest-neighbor interactions giving rise to a staircase of phases. Again, we focus on the effects of the untruncated long-range interactions.

Response: Anonymous Report 2 on 2024-2-26 (Invited Report)

We thank the referee for the report. We have modified the text accordingly in order to address the referees remarks and criticisms.

Below we address the points raised by the referee.

i) Regarding the nearest-neighbour case: the literature states that quantum fluctuations have a larger impact on systems with geometrical frustration. The model on the triangular lattice is subject to geometrical frustration, therefore a larger impact of quantum fluctuation on the phase diagram is expected. We have added a clarification of this point in Sec. 5.

ii) The classical approximation captures the transition at the Heisenberg point correctly since the transition is driven by the change in symmetry of the Hamiltonian from an easy-axis to a rotationally invariant interaction. This change in symmetry is present in the classical spin picture, as well as, the full quantum mechanical problem for the same parameter value $t/V=1/2$. We have added a clarification of this point in Sec. 5.

iii) We have strengthened the emphasis on the quantum phases in the results section. We realised that we used the misleading name "order" for "supersolid" phases in Figs. 5, 6, 10. We correspondingly modified it in the figure captions and in the text. Further, we added text passages discussing the different supersolid phases for the square and the triangular lattice.

In Fig. 7, we have added real and momentum space depictions of the complex supersolid phases on the square lattice. We have added a picture how to understand the supersolid phases on the square lattice in terms of defective checkerboard patterns.

Regarding the triangular lattice, we are grateful for the reference to the quantum Monte Carlo study PRL 104, 125302 (2010). We have added a discussion of the reference in comparison to our results in Sec. 7. In the conclusion we highlighted the application of the discussed method to determine relevant observables and unit cells for further numerical studies.

---

## Round 2 · List of Changes

Warnings issued while processing user-supplied markup:

  • Inconsistency: plain/Markdown and reStructuredText syntaxes are mixed. Markdown will be used.
    Add "#coerce:reST" or "#coerce:plain" as the first line of your text to force reStructuredText or no markup.
    You may also contact the helpdesk if the formatting is incorrect and you are unable to edit your text.

List of changes

line 19 "square lattice" -> "square and triangular lattice"

line 55 "important" -> "prominent"

line 56 "Due" -> "Thanks"

line 57 "allow for applications as quantum simulators of" -> "permit to shed light on"

line 58 "of" -> "on the predictions of"

line 79 added "three lattice geometries often discussed in the literature:"

line 82 to 99 paragraph rephrased

line 109 "In the following, we introduce the model in Sec. 2 in the particle as well as the spin language." -> "This paper is organised as follows. In Sec. 2 we introduce the model in the particle as well as the spin language."

line 112 added "In Sec. 2.3 we define the three lattices geometries that are considered in this work (square, honeycomb and triangular lattice)." due to the shift of the paragraph

line 115 "We further discuss how we characterise ground states from the results of the mean-field calculation, as well as the inherent limitations of the approach." -> "We then detail how we characterise ground states from the results of the mean-field calculation, and critically discuss the inherent limitations of the approach."

line 122 "discuss" -> "analyse"

line 128 "investigate" -> "consider"

line 129 "and in a grand-canonical ensemble" -> "with a variable particle number"

line 133 to 136 paragraph rephrased

line 140 "From former studies in the atomic limit without hopping on a one-dimensional chain [61,62] and two-dimensional lattices [34] a devil’s staircase of crystalline phases is expected considering the full long-range interaction." -> A former study analysing the full long-range interations in the atomic limit t = 0 predicts a devil staircase of crystalline phases [34].

line 146 added "By"

line 148 "where" -> "Here,"

line 149 to 152 paragraph rephrased

line 155 "for any value of the exponent $\alpha$"

line 163 "and the absence of any longitudinal field at the symmetry line where" -> "about the symmetry line where the longitudinal field vanishes,"

Sec. 2.3 is former Sec 3

line 211 "superfluid" -> "superfluid phase"

line 213 "the long-range" -> "the corresponding order by long-range"

Eq. (9) indices changed

line 225 and 227 $f=n/m$ corrected

line 230 "unit cells with more than one site" -> "composed by multiple sites"

line 231 "against finite" -> "against quantum fluctuations at finite"

line 242 added "In order to identify supersolids with a complex sublattice structure, a careful choice of an order parameter is required, see Sec. 5.2.1."

Caption Table 1 "momentum" -> "wave-vector"

Caption Table 1 "dependent" -> "determined by"

Caption Table 1 "complex sublattice structure" -> "with more than two sublattices"

line 266 "pattern" -> "pattern minimizing energy"

line 297 "In addition" -> "In addition,"

Caption Fig. 2 "second (first)" -> "first (second)" - statement corrected

Caption Fig.2 "illustrate" -> "display"

line 383 "The phase transition between phases is first-order, if the two phases have a different diagonal long-range order." -> "First-order phase transitions separate two phases with different diagonal long-range order."

line 406 added a paragraph on the Heisenberg point

line 415 added a paragraph on the impact of frustration

line 430 "present" -> "analyse"

line 432 "The classical spin approach ground states ... resulting from the numerical optimization procedure, which were used to create Figs. 5-7, can be found in Ref. [79]." -> "We refer to Ref. [64] for the data on the classical spin approach ground states ... resulting from the numerical optimization procedure. This data was used to create Figs. 5-8."

line 457 "in some" -> "even though in some"

line 466 "discuss" -> "analyse"

Fig. 5 has changed. The misleading term "order" has been replaced by the more accurate term "supersolid".

Fig. 6 has changed. The misleading term "order" has been replaced by the more accurate term "supersolid". The translational vectors depicted for the "5/14 solid" and "4/7 - 1/14 - 5/14 supersolid" have been wrong and were replaced by the correct ones.

Fig. 7 has been added.

line 503 to 516 paragraphs rephrased

line 542 "In comparison" -> "In contrast to"

line 544 to 557 paragraph rephrased

line 577 to 582 paragraph rephrased

Fig. 10 has changed. The misleading term "order" has been replaced by the more accurate term "supersolid".

line 593 to 595 rephrased and shortened paragraph

line 596 to 626 added a paragraph discussing the reference L. Pollet et al. [36] (2010)

Sec. 7 Conclusion has been reformulated extensively

line 683 "KPS acknowledges" -> "KPS and JAK acknowledge"

The reference Su2023 has been updated from the arXiv to the published version.

The reference Adelhardt2024Review was added.

The reference Koziol2024 was added.

The reference Lan2018 added.

The reference Pollet2007 was added.

The reference Pollet2010 was added.

The reference Prokofev1998 was added.

The reference Suthar2020 was added.

The reference Zhang2022 was added.

---

## Round 3 · List of Changes

* * *
List of changes
* * *
- Sec. 3.1: We added an entire paragraph at the end of the section outlining a novel numerical tool to efficiently evaluate the resummed couplings.

- Sec. 3.4: We added two statements, the first emphasising the comparison to quantum Monte Carlo results, the second summarising general statements about quantum fluctuations.

- Sec. 7: We added the second paragraph, where we outline how ultracold dipolar atoms or molecules can be used to realize our model. We summarize how to experimentally tune model parameters and how to experimentally probe diagonal and off-diagonal long-range order in the model.

- Sec. 7: We added the second to last paragraph, where we elaborate on potential improvements of the approach.

- Sec. 8: We adjusted the acknowledgements statement.

- Bibliography: We added the reference Epstein1903

- Bibliography: We added the reference Epstein1906

- Bibliography: We added the reference Crandall2012

- Bibliography: We added the reference Buchheit2021

- Bibliography: We added the reference Buchheit2022

- Bibliography: We added the reference Buchheit2024

- Bibliography: We added the reference KoziolDataOld

- Bibliography: We added the reference Buchheit2024Code

- Bibliography: We added the reference Griesmaier2005

- Bibliography: We added the reference Lu2011

- Bibliography: We added the reference Aikawa2012

- Bibliography: We added the reference Ni2008

- Bibliography: We added the reference Danzl2008

- Bibliography: We added the reference Deiglmayr2008

- Bibliography: We added the reference dePaz2013

---

## Editorial Decision

accepted_in_target_journal